# Acquired miR-142 deficit in leukemic stem cells suffices to drive chronic myeloid leukemia into blast crisis

Bin Zhang [1,13,14] ✉, Dandan Zhao[1,13], Fang Chen[1,13], David Frankhouser [2], Huafeng Wang[1,3], Khyatiben V. Pathak[4,5], Lei Dong[6], Anakaren Torres[4,5], Krystine Garcia-Mansfield [4,5], Yi Zhang[1,3], Dinh Hoa Hoang[1], Min-Hsuan Chen[7], Shu Tao [7], Hyejin Cho [7], Yong Liang [8], Danilo Perrotti[9,10], Sergio Branciamore[2], Russell Rockne [2], Xiwei Wu[7], Lucy Ghoda[1], Ling Li [1], Jie Jin[3], Jianjun Chen [6], Jianhua Yu [11], Michael A. Caligiuri [11], Ya-Huei Kuo [1], Mark Boldin[6], Rui Su [6], Piotr Swiderski[8], Marcin Kortylewski [12], Patrick Pirrotte [4,5], Le Xuan Truong Nguyen[1,4,14] ✉ & Guido Marcucci [1,14] ✉

The mechanisms underlying the transformation of chronic myeloid leukemia (CML) from chronic phase (CP) to blast crisis (BC) are not fully elucidated. Here, we show lower levels of miR-142 in CD34+CD38− blasts from BC CML patients than in those from CP CML patients, suggesting that miR-142 deficit is implicated in BC evolution. Thus, we create miR-142 knockout CML (i.e., *miR-142−/− BCR-ABL*) mice, which develop BC and die sooner than miR-142 wt CML (i.e., *miR-142+/+ BCR-ABL*) mice, which instead remain in CP CML. Leukemic stem cells (LSCs) from *miR-142−/− BCR-ABL* mice recapitulate the BC phenotype in congenic recipients, supporting LSC transformation by miR-142 deficit. State-transition and mutual information analyses of "bulk" and single cell RNA-seq data, metabolomic profiling and functional metabolic assays identify enhanced fatty acid β-oxidation, oxidative phosphorylation and mitochondrial fusion in LSCs as key steps in miR-142-driven BC evolution. A synthetic CpG-miR-142 mimic oligodeoxynucleotide rescues the BC phenotype in *miR-142−/− BCR-ABL* mice and patient-derived xenografts.

Chronic myeloid leukemia (CML) is a myeloproliferative disorder characterized by the Philadelphia chromosome, a translocation of chromosomes 9q34 and 22q11 resulting in a fusion oncogene, *BCR−ABL1* that encodes a constitutively activated tyrosine kinase (TK) mutant. *BCR−ABL1* transforms hematopoietic stem cells (HSCs) into leukemic stem cells (LSCs)[1]. The disease usually presents in a chronic phase (CP) characterized by myeloproliferation and, if left untreated, progresses to an accelerated phase (AP) and eventually to blast crisis (BC). Even though nowadays, CP CML can be effectively treated with TK inhibitors (TKIs), these patients are often committed to a life-long treatment since otherwise persistent LSCs may drive disease relapse

and possibly BC transformation. Unfortunately, once progressed to AP/BC, these patients require allogeneic stem cell transplant (alloSCT) to achieve a cure[2]. The underlying molecular mechanisms of the disease evolution from CP to AP/BC are complex, multifaceted, and remain to be fully understood[3].

LSCs are primitive leukemic cells capable of indefinite self-renewal and of initiating and maintaining leukemia growth (i.e., "stemness")[4,5]. They are few, sparse, and difficult to identify in the bone marrow (BM) or other hematopoietic organs. In fact, not only do they share common membrane antigens with normal hematopoietic stem and progenitor cells (HSPCs), but LSCs also reside in immunophenotypically diverse

leukemic cell subpopulations. To date, methods for recognition and characterization of LSCs remain mainly functional (e.g., colony formation in vitro and disease initiation in recipient mice in vivo). Quiescence, high dependence on oxidative phosphorylation (OxPhos), and low production of reactive oxygen species (ROS) are "key" features to distinguish LSCs from normal HSCs both in acute myeloid leukemia (AML) and CML[5–8]. A recent report also highlighted genomic differences between CP- and BC-LSCs, suggesting that disease transformation may depend on the functional evolution of LSCs[9].

MicroRNAs (miRNAs) are short non-coding RNA molecules that downregulate target messenger (m)RNAs and in turn, their encoded proteins. MiRNAs have been found to be involved in multiple homeostatic and functional processes of normal cells, and their aberrant up and downregulation in cancer have been associated respectively with downregulation of tumor suppressor genes and upregulation of oncogenes. *MIR142*, located at chromosome band 17q22, is initially transcribed into a primary (pri)-miR-142, and eventually matures into two distinct miRNAs, miR-142-3p and miR-142-5p (hereafter collectively referred to as miR-142)[10]. MiR-142 is dynamically expressed and reportedly plays a regulatory role in hematopoiesis and immunity[11]. In fact, miR-142 is involved in hemangioblast differentiation and HSC development as well as in differentiation and/or activation of T lymphocytes, natural killer cells (NK) and dendritic cells (DC)[12–17]. Loss of the *mir142* gene in the mouse results in expansion of BM HSPCs, decreased hematopoietic output, and reduction of T, B and NK cells[12].

In humans, loss of miR-142 function has been reportedly associated with blood cancers. Mutations and/or downregulation of miR-142 have been found in follicular and diffuse large B cell lymphoma and acute lymphocytic leukemia (ALL)[18]. More recently, *MIR142* was found to be mutated or downregulated in AML[19,20] and cooperates with other gene mutations[21] to promote leukemia growth via *ASH1L*-dependent *HOXA9* and *HOXA10* upregulation[22]. In myeloproliferative neoplasms, mutations of miR-142 have not been commonly found[23]. Nevertheless, preliminary data suggest that downregulation of miR-142 activates antiapoptotic mechanisms[24] and predicts for TKI resistance in CML[25].

Herein we show that miR-142 deficit occurs in BC CML patients and drives disease by enhancing the bioenergetic oxidative metabolism and mitochondrial fusion in LSCs in the absence of any additionally acquired gene mutations. Conversely, correcting the miR-142 deficit with a novel synthetic miR-142 mimic oligodeoxynucleotide (ODN) decreased fatty acid β-oxidation (FAO) and OxPhos and induced mitochondrial fragmentation, thereby rescuing the phenotype and significantly prolonged survival of BC genetically engineered mouse models (GEMMs) and patient-derived xenografts (PDXs). Taken altogether, our results demonstrate that a miR-142 deficit suffices in transforming CP-LSC into BC-LSCs and represents a druggable target for disease elimination.

## Results

### miR-142 deficit induces BC-like transformation in the CP CML mouse

In search of new therapeutic targets for BC CML, we observed that miR-142 levels were significantly lower in mononuclear cells (MNCs), CD34+ HSPCs, and CD34+CD38− HSCs from BC CML patients compared with counterparts from healthy donors [HSCs: 4.2-fold lower ($p = 0.002$); Fig. 1a] or from CP CML patients [HSCs: 3.75-fold lower ($p < 0.0001$); Fig. 1a]. Hence, considering the known role of miR-142 in hematopoiesis, we postulated that an acquired miR-142 deficit has a mechanistic role in the BC transformation of CML.

To test this hypothesis, we initially turned to GEMMs. The *SCLtTA/BCR-ABL* (hereafter referred to as *BCR-ABL*) mouse in B6 background is a CP CML model that expresses *BCR-ABL* under the *SCL* promoter upon tetracycline withdrawal (i.e., tet-off)[26–28]. Different from the human disease, CP CML in these mice does not spontaneously evolve into BC

CML. Thus, to assess the impact of an acquired miR-142 deficit on the disease evolution, we crossed the miR-142 wt *BCR-ABL* B6 mice with miR-142 knock-out (KO) (*miR-142−/−*) B6 mice[12].

The phenotype of the *miR-142−/−* mice has been previously reported[12]. In our hands, these mice had minimal or no miR-142 expression (Fig. 1b), presented with no internal organ defects except for splenomegaly, and developed a significant T lymphopenia. In the normal mouse, BM Lineage (Lin)−Sca-1+c-Kit+ (LSK) cells comprise multiple HSPC subsets, including long-term engrafting hematopoietic stem cells (LT-HSC, Flt3−CD150+CD48− LSK) that correspond to human HSCs, and other short-term engrafting multipotent progenitors (MPP) [i.e., Flt3−CD150−CD48− (MPP1), Flt3−CD150+CD48+ (MPP2), Flt3−CD150−CD48+ (MPP3) and Flt3+CD150− lymphoid-primed MPP (LMPP)][27,29]. To this end, *miR-142−/−* mice had a prominent BM and spleen expansion of LSKs and myeloid precursors [i.e., MPP3 and granulocyte-macrophage progenitors (GMP, Lin−Sca-1−c-Kit+CD34+FcγRII/IIIhi)] (see Supplementary Fig. 1a–c; left two panels; Supplementary Table 1). Importantly, none of the *miR-142−/−* mice developed leukemia over time (Supplementary Fig. 2a–c).

Crossing the *BCR-ABL* mice with the *miR-142−/−* mice, we obtained homozygous (homo, i.e., *miR-142−/−BCR-ABL*) and heterozygous (het, i.e., *miR-142+/−BCR-ABL*) miR-142 KO *BCR-ABL* progeny (Fig. 1c). Of note, even though higher expression levels of *SCL* and *BCR-ABL* are not associated with BC transformation in the *BCR-ABL* mice, they may greatly enhance myeloproliferation and survival rates. Thus, for a meaningful phenotypic comparison, we matched *miR-142−/−BCR-ABL*, *miR-142+/−BCR-ABL* and *miR-142+/+BCR-ABL* mice not only for age and gender, but also for *SCL* and *BCR-ABL* levels (by Q-RT-PCR). Upon *BCR-ABL* induction, *miR-142+/+BCR-ABL* mice developed CP CML and died of excessive myeloproliferation without evidence of AP/BC transformation (Supplementary Fig. 2c; the third panel). In contrast, both *miR-142+/−BCR-ABL* and *miR-142−/−BCR-ABL* mice had 100% penetrance of a BC-like phenotype characterized by higher white blood cell (WBC) counts (Fig. 1d; Supplementary Fig. 2a), circulating blasts (Fig. 1e, Supplementary Fig. 2c, d), anemia and thrombocytopenia (Supplementary Fig. 2e) as compared with the *miR-142+/+BCR-ABL* mice. The BM of miR-142−/−BCR-ABL mice presented with lower LT-HSC and megakaryocyte-erythrocyte progenitors (MEP, Lin−Sca-1−c-Kit+CD34low+FcγRII/IIIlow/−), and higher LSK, MPP3, GMP, and blasts (Fig. 1f; Supplementary Fig. 1a, b, right two panels; summarized in Supplementary Table 1). Furthermore, the *miR-142−/−BCR-ABL* mice had significantly larger spleens with extramedullary hematopoiesis characterized by expansion of LSKs and other myeloid progenitors and by a significant decrease of T cells compared with the *miR-142+/+BCR-ABL* controls (Fig. 1g; Supplementary Figs. 1c and 2a, right two panels; Supplementary Table 1). Both *miR-142+/−BCR-ABL* and *miR-142−/−BCR-ABL* mice had a shorter survival compared with the *miR-142+/+BCR-ABL* mice (median survival: homo 32 vs het 45 vs wt 55 days; $p < 0.0001$ for homo vs wt; $p = 0.006$ for het vs wt; $p = 0.004$ for homo vs het; Fig. 1h), suggesting a dosage effect of miR-142 deficit on the aggressiveness of the transformed phenotype. Comparing the BM and spleen cellular subpopulations from leukemic *miR-142+/+BCR-ABL* and *142−/−BCR-ABL* mice with those of the respective age and gender-matched normal (non-leukemic) *miR-142+/+* or *miR-142−/−* controls, we noted that, while some of the changes were likely driven solely by the miR-142 deficit [e.g., BM: MPP1 and common lymphoid progenitors (CLP); spleen: erythrocyte] or the BCR/ABL expression (e.g., BM: CMP and B cells), the majority of them were the results of the combinatorial effects of both "hits" (e.g., BM: LSK, LT-HSC, MPP2, MPP3, LMPP, GMP, MEP, myeloid, T and erythrocyte; spleen: LSK, LT-HSC, MPP1, MPP2, MPP3, LMPP, CMP, GMP, MEP, myeloid, B and T cells) (Supplementary Fig. 1b, c).

It has been reported that development of BC phenotype is associated with upregulation of Musashi-2 (MSI2), an RNA-binding protein involved in stem cell self-renewal and differentiation[30–34]. To

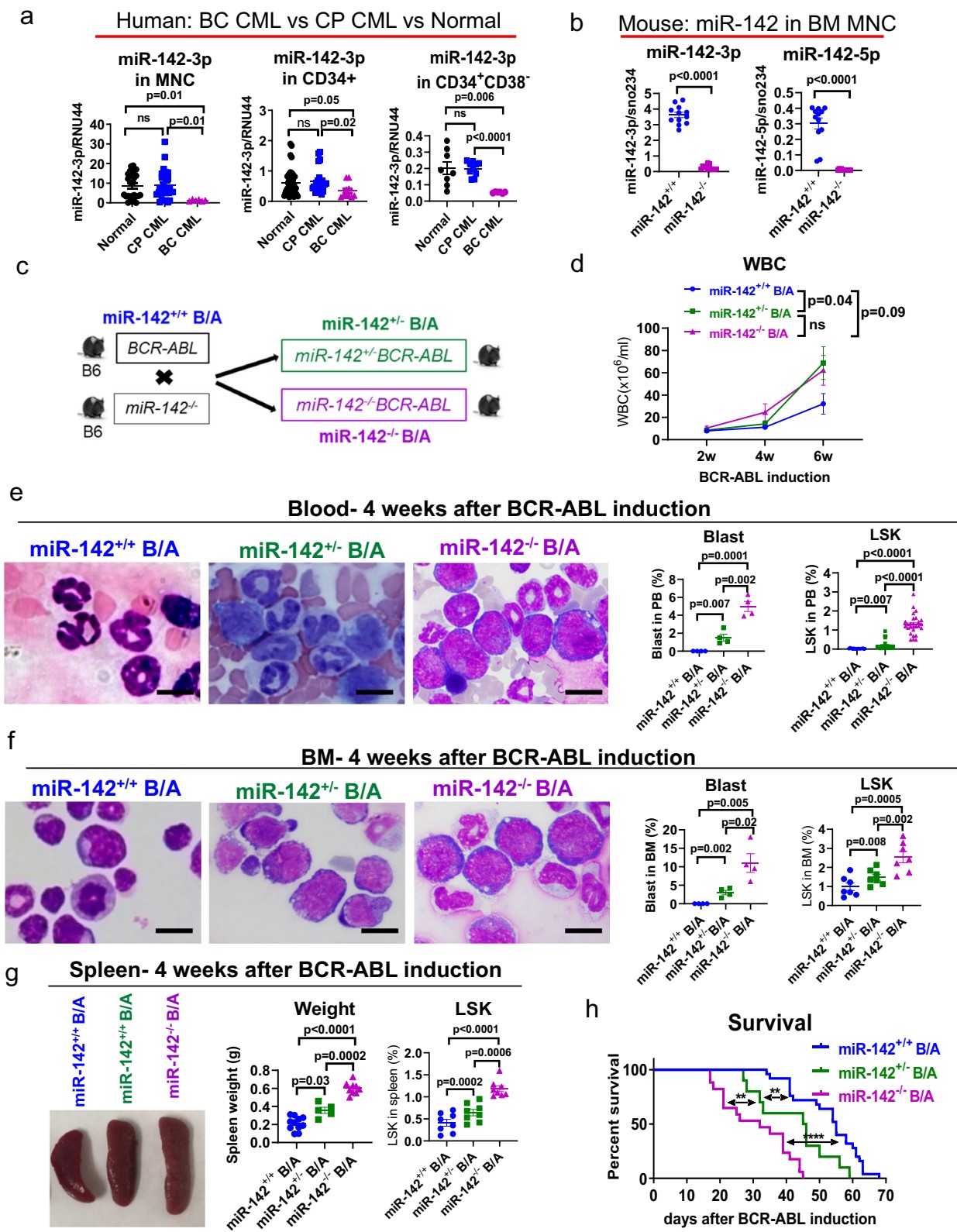

this end, we observed higher Msi2 mRNA and protein levels in LSKs from *miR-142⁻/⁻BCR-ABL* (BC CML) mice compared with those from *miR-142⁺/⁺BCR-ABL* (CP CML) mice (Supplementary Fig. 3a, b). Interestingly, we also observed higher Msi2 mRNA and protein levels in LSKs from the *miR-142⁻/⁻* mice compared with those from the *miR-142⁺/⁺* mice (Supplementary Fig. 3c, d), suggesting a functional interplay between miR-142 and Msi2. Accordingly, downregulation of miR-142 in miR-142⁺/⁺BCR-ABL LSKs using a miR-142 inhibitor (CpG-anti-miR-142 ODN) was associated with increased Msi2 mRNA and protein levels (Supplementary Fig. 3e, f). Conversely, upregulation of miR-142 levels in miR-142⁻/⁻BCR-ABL LSKs with a miR-142 mimic (CpG-M-miR-142; see methods for details) reduced Msi2 mRNA and protein levels (Supplementary Fig. 3g, h). Mining TargetScan (8.0) and miRDB databases, MSI2 was identified as a predicted target of

**Fig. 1 | miR-142 deficit induces BC-like transformation in the CP CML mouse.**
**a** MiR-142 levels in bone marrow (BM) mononuclear cells (MNCs), CD34+ hematopoietic stem and progenitor cells (HSPCs), or CD34+CD38− hematopoietic stem cells (HSCs) from healthy individuals (normal; n = 14 biologically independent samples for MNCs and n = 8 for CD34+ and CD34+CD38−), chronic phase (CP) CML patients (n = 17 biologically independent samples for MNCs and n = 10 for CD34+ and CD34+CD38−), or blast crisis (BC) CML patients (n = 6 biologically independent samples for all three cell populations), analyzed by Q-RT-PCR. **b** MiR-142 levels in BM MNCs from *miR-142−/−* and *miR-142+/+* mice (n = 12 each), analyzed by Q-RT-PCR. **c** Schematic design of the mouse crossings. Upon tetracycline withdrawal to induce BCR-ABL expression, *miR-142−/−BCR-ABL*, *miR-142+/−BCR-ABL* and *miR-142+/+BCR-ABL* mice were monitored for **d** white blood cell (WBC) counts (n = 20, 13 and 15 mice respectively) every two weeks, **e** blood (n = 4 mice per strain for blasts and n = 18 mice per strain for LSK) and **f** BM leukemic blasts (n = 4 mice per strain for blasts

and n = 7 mice per strain for LSK) assessed by microscopy (left, scale bar: 10 μM) and Lin−Sca-1+c-Kit+ (LSKs) by flow cytometry (right), **g** spleen size and weight (left) and LSKs by flow cytometry (right) four weeks post BCR-ABL induction (n = 8 mice per strain). **h** Survival of *miR-142−/−BCR-ABL* (n = 17), *miR−142+/−BCR-ABL* (n = 10) and *miR-142+/+BCR-ABL* (n = 25) mice after BCR-ABL induction (Log-rank test, *miR−142−/−BCR-ABL* vs *miR-142+/−BCR-ABL*: p = 0.004; *miR-142+/−BCR-ABL* vs *miR-142+/+BCR-ABL*: p = 0.007; *miR-142−/−BCR-ABL* vs *miR-142+/+BCR-ABL*: p < 0.0001). For **a−h**, source data are provided as a Source Data file. For **e** and **f**, results from one of the three independent experiments are shown (n = 3). Abbreviation: miR-142+/+B/A: *miR-142+/+BCR-ABL*; miR-142−/−B/A: *miR-142−/−BCR-ABL*; CP: chronic phase; BC: blast crisis. Comparison between groups was performed by two-tailed, unpaired t-test. Results shown represent mean ± standard error of the mean (SEM). Significance values: **p < 0.01; ****p < 0.0001; ns, not significant.

miR-142-3p as supported by a conserved 8mer miR-142-3p-binding site in the 3′ UTR of MSI2 (Supplementary Fig. 3i). Thus, these results suggested that upregulation of Msi2 in these models was due to the absence of its targeting miRNA (i.e., miR-142). Of note, mining publicly available MSI2 CLIP datasets[35–37], we found no evidence of miR-142 among the identified MSI2 binding targets.

### miR-142 promotes evolution of CP-LSCs into BC-LSCs

In the *miR-142+/+BCR-ABL* mouse, LSKs are reportedly LSC-enriched[27]. However, since we observed circulating *miR-142−/−BCR-ABL* blasts, immunophenotypically consistent with LSKs and GMPs, and previous reports suggested that LSCs may also reside in the GMP fraction of CML models[38], we conducted transplant experiments to identify which of these fractions from *miR-142−/−BCR-ABL* or *miR-142+/+BCR-ABL* mice were more LSC-enriched. LSKs and GMPs from CD45.2 *miR-142−/−BCR-ABL* or *miR-142+/+BCR-ABL* mice were transplanted into normal congenic CD45.1 wt recipients (Fig. 2a). While we observed robust long-term engraftment in the recipient mice with as few as 2,000 LSKs from *miR-142+/+BCR-ABL* or *miR-142−/−BCR-ABL* mice (Fig. 2b), no long-term engraftment was observed with as many as 20,000 GMPs from *miR-142+/+BCR-ABL* or *miR-142−/−BCR-ABL* mice (Fig. 2c). Importantly, when engrafted into congenic normal wt CD45.1 recipients, LSKs from both *miR-142−/−BCR-ABL* and *miR-142+/−BCR-ABL* mice (CD45.2) recapitulated the BC phenotype whereas LSKs from *miR-142+/+BCR-ABL* donors recapitulated the CP phenotype (Fig. 2d, e), suggesting that miR-142 deficit had stably transformed CP-LSCs into BC-LSCs. To test if secondary genomic "hits" (i.e., gene mutations) cooperated with miR-142 deficit in inducing BC transformation, we subjected BM LSKs harvested from *miR-142−/−BCR-ABL* at the time of first appearance of circulating blasts (3–4 weeks after BCR-ABL induction) and those from *miR-142+/+BCR-ABL* mice, to whole genome sequencing. While we found the expected mir142 gene deletion (Supplementary Fig. 4a), we did not observe additional gene copy number variations or newly acquired mutations in the *miR-142−/−BCR-ABL* LSKs as compared with the *miR-142+/+BCR-ABL* LSKs.

At the transcriptome level, LSKs constitute a heterogeneous cell population, comprised of cell fractions that are seemingly primed for further HSPC-lineage differentiation[39]. Thus, to assess if the miR-142 deficit affected the BM LSC-enriched LSK landscape, we utilized single cell (sc) RNA-seq analysis of BM *miR-142+/+BCR-ABL* and *miR-142−/−BCR-ABL* LSKs which were harvested from the respective mice 2 weeks after BCR-ABL induction (Fig. 2f). Using the Leiden clustering algorithm to identify groups of cells with similar transcriptomes, we identified 8 distinct LSK clusters (Fig. 2g, h). We annotated these clusters into distinct lineage-primed subsets using the expression levels of hematopoietic gene transcription factors and cluster differentiation (CD) antigens (Supplementary Fig. 4b, c). Using Elane and FLT3 expression, we dichotomized LSKs into myeloid-primed and lymphoid-primed cells respectively[40,41]. Consistent with enriched downstream myeloid progenitors by flow cytometry

(Supplementary Fig. 1a, b), we observed expansion of the Elane-expressing myeloid/neutrophil-primed (cluster 2) and myeloid/lymphoid-primed (cluster 5) LSKs, and reduction of FLT3-expressing lymphoid-primed (cluster 1) in the *miR-142−/−BCR-ABL* LSKs compared with *miR-142+/+BCR-ABL* LSKs (Fig. 2g, h). The non-lineage-primed LSKs (cluster 3), likely representing LT-HSC, were significantly decreased as also observed by flow cytometry analysis (Supplementary Fig. 1a, b). Interestingly, gene set enrichment analysis (GSEA) revealed significant increases in the expression levels of genes involved in OxPhos and lipid metabolism in addition to those involved in cell cycle and inflammation in the *miR-142−/−BCR-ABL* LSK clusters compared with the *miR-142+/+BCR-ABL* LSK clusters (Supplementary Fig. 5a−c).

Human CP CML samples engraft poorly in the mouse, while human BC CML samples have enhanced engraftment capacity[5]. Thus, to assess the relevance of the miR-142 deficit to "stemness" (i.e., engrafting and disease initiating capacities) in the human disease as we observed in the mouse, we knocked-down (KD) miR-142 in LSC-enriched CD34+ cells from CP CML patients, by transducing a GFP-expressing (GFP+) anti-miR-142 lentivirus vector (Supplementary Fig. 5d, e). In vitro, GFP+ miR-142 KD CD34+ cells had reduced spontaneous apoptosis and increased cell growth rates (Supplementary Fig. 5f) compared with controls (i.e., nontarget control vector-transduced CD34+ cells hereafter called GFP+ miR-142 wt CD34+ controls). In vivo, when engrafted into immunodeficient NSG mice, GFP+ miR-142 KD CD34+ CML cells produced higher blood and BM engraftment rates (i.e., human CD45+GFP+ cells) as compared with GFP+ miR-142 wt CD34+ controls (2.8% vs 0.6% in BM four weeks after transplantation, p < 0.0001; Supplementary Fig. 5g), thereby supporting that increased "stemness" associated with the miR-142 deficit.

### State-transition from CP-LSCs to BC-LSCs is captured by changes in the expression of genes involved in bioenergetic oxidative metabolism

To gain further insight into how miR-142 deficit drives a BC-like transformation and identify pathways and targets mediating the disease transformation, we performed bulk RNA-seq on LSKs harvested from normal *miR-142+/+* (wt) and *miR-142−/−* (miR-142 KO) mice as well as from leukemic *miR-142+/+BCR-ABL* (CP CML) and *miR-142−/−BCR-ABL* (BC CML) mice, two weeks after BCR-ABL induction (Fig. 3a). Focusing on *miR-142−/−BCR-ABL* vs *miR-142+/+BCR-ABL* LSKs, we identified 506 differentially expressed genes (DEGs), 211 upregulated and 295 downregulated (Fig. 3b). GSEA of the Hallmark gene sets (MSigDB) identified metabolism-related gene sets, i.e., OxPhos (M5936), protein secretion (M5935), adipogenesis (M5905), and glycolysis (M5937) [hereafter referred to as enriched metabolism Hallmark gene sets (EMHGS); Fig. 3c; Supplementary Fig. 6a−d] as the top four enriched gene sets in *miR-142−/−BCR-ABL* LSKs. Of note, these four gene sets were also among the top ranked enhanced gene sets in the normal (i.e., non-leukemic) *miR-142−/−* vs *miR-142+/+* LSKs (Fig. 3d).

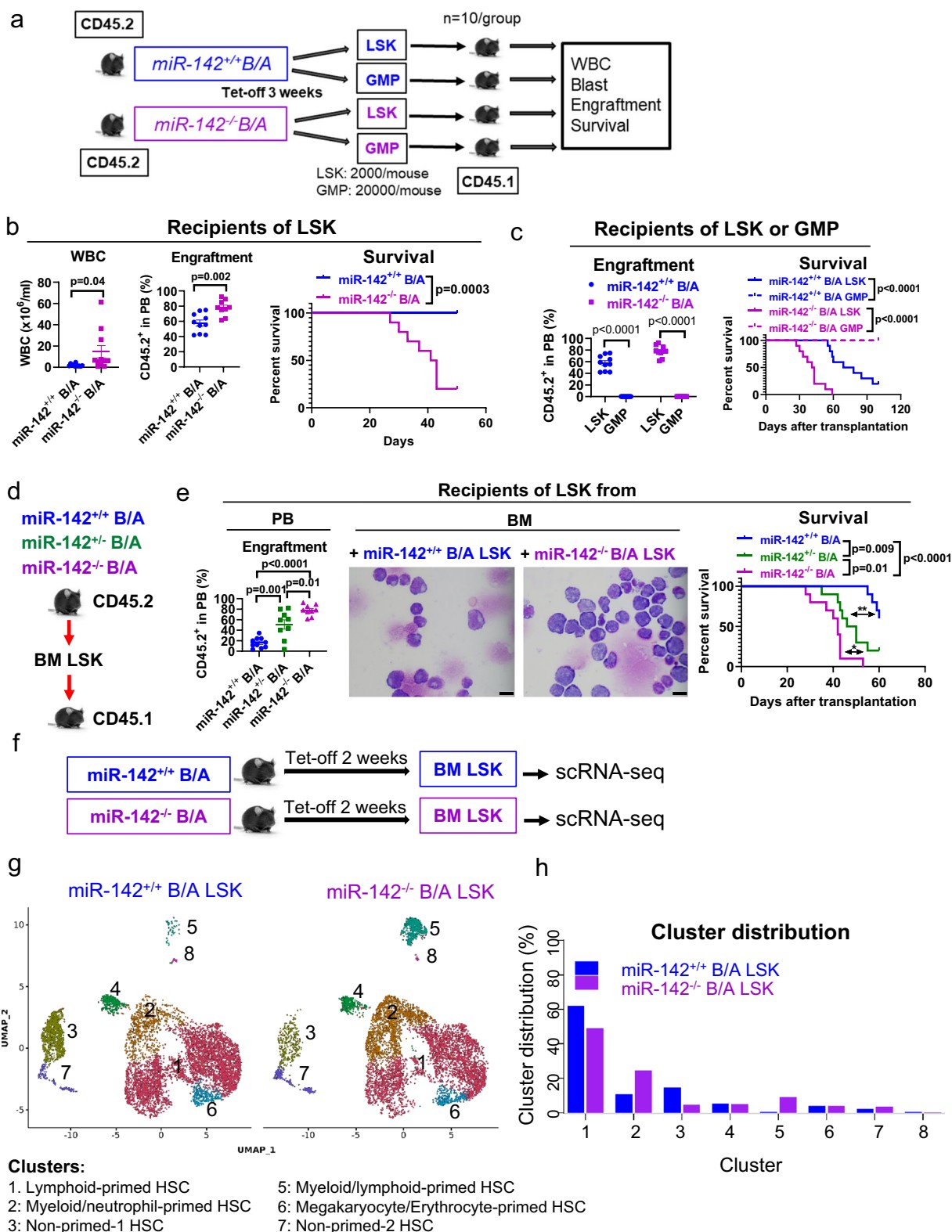

**Clusters:**
1. Lymphoid-primed HSC
2. Myeloid/neutrophil-primed HSC
3. Non-primed-1 HSC
4. Myeloid-primed HSC
5. Myeloid/lymphoid-primed HSC
6. Megakaryocyte/Erythrocyte-primed HSC
7. Non-primed-2 HSC
8. Eosinophil-primed HSC

To assess the relevance of the EMHGS changes to the BC transformation, we initially pursued an analytical systems biology approach based on state-transition analysis. We have previously shown that the evolution from health to leukemia can be modeled in the mouse as a state-transition of the transcriptome across a state-space constructed using data from gene expression principal component (PC) analysis[42,43]. Thus, using all the RNA-seq data, we constructed a "whole transcriptome" state-space and mapped the state-transition from normal LSKs to CP-LSKs and in turn to BC-LSKs. Among the different PCs, PC1 and PC2 best encoded, respectively, the information that separated LSKs according to their leukemic (i.e., normal vs CP and BC) and miR-142 (i.e., wt vs KO) states (Fig. 3e). To test if the EMHGS also contained information that, similar to the whole transcriptome state-space, could identify the distinct LSK states, we constructed an EMHGS

**Fig. 2 | Identification and single cell gene expression profiles of BC-LSCs vs CP-LSCs. a–c** Schematic design and results of the experiment. **a** LSK (2000/mouse, n = 10 mice per group) and GMP (20,000/mouse, n = 10 mice per group) populations from CD45.2 *miR-142⁻/⁻BCR-ABL* and *miR-142⁺/⁺BCR-ABL* mice (BCR-ABL were induced for 3 weeks) were transplanted into congenic CD45.1 wt recipients. **b** White blood cell (WBC) counts (p = 0.04), peripheral blood (PB) engraftment rates (CD45.2⁺, p = 0.002) by flow cytometry measured at four weeks after transplantation, and survival (Log-rank test, p = 0.001) of recipients transplanted with LSKs from *miR-142⁻/⁻BCR-ABL* or *miR-142⁺/⁺BCR-ABL* mice (n = 10 mice per strain) are shown. **c** PB engraftment rates (CD45.2⁺, p < 0.0001) at four weeks post transplantation and survival (Log-rank test, p < 0.0001) of recipients transplanted with LSKs or GMPs from *miR-142⁺/⁺BCR-ABL* and *miR-142⁻/⁻BCR-ABL* mice (n = 10 per group) are shown. **d, e** Schematic design and results of the experiment. **d** LSKs from CD45.2 *miR-142⁻/⁻BCR-ABL*, *miR-142⁺/⁻BCR-ABL* and *miR-142⁺/⁺BCR-ABL* mice (BCR-ABL were induced for 3 weeks) were transplanted into congenic CD45.1 wt recipients. **e** PB engraftment rates at 4 weeks post transplantation, BM blasts assessed by microscopy (scale bar: 10 μM), and survival of these recipient mice are shown (n = 10 mice per group; Log-rank test, *miR-142⁻/⁻B/A* vs *miR-142⁺/⁺B/A*: p = 0.01; *miR-142⁺/⁻B/A* vs *miR-142⁺/⁺B/A*: p = 0.009; *miR-142⁻/⁻B/A* vs *miR-142⁺/⁺B/A*: p < 0.0001). For **b, c, e, h**, source data are provided as a Source Data file. For **e**, results from one of the three independent experiments are shown (n = 3). For **b, c, e**, comparison between groups was performed by two-tailed, unpaired t-test. Results shown represent mean ± SEM. Significance values: *p < 0.05; **p < 0.01. **f–h** Schematic design and results of the experiment. **f** BM LSK cells from *miR-142⁻/⁻BCR-ABL* and *miR-142⁺/⁺BCR-ABL* mice (BCR-ABL were induced for two weeks) were sorted for scRNA-seq. **g** LSK clusters and **h** cluster cell distribution are shown. B/A BCR-ABL, tet-off tetracycline withdrawal, GMP granulocyte-macrophage progenitors, PB peripheral blood, BM bone marrow, scRNA-seq single cell RNA sequencing, SEM standard error of the mean.

state-space using only the 655 genes included in the EMHGS (Fig. 3f). Like the whole transcriptome state-space, PC1 and PC2 of the EMHGS state-space correctly mapped the LSK populations according to their leukemic and miR-142 states (Fig. 3f). To this end, PC1 and PC2 of the EMHGS state-space were significantly correlated with PC1 and PC2 of the whole transcriptome state-space (PC1s, $R^2$ = 0.90; PC2s, $R^2$ = 0.91; p < 0.01; Fig. 3g).

While these results supported that changes in EMHGS sufficed to inform on the LSK state, they did not provide much insight into how much information was provided by the EMHGS versus (vs) how much by the "remaining" transcriptome (i.e., whole transcriptome minus EMHGS). To this end, we then conducted a mutual information (MI) analysis[44]. MI can be thought of as a generalization of correlation analysis, which also captures non-linear relationships between variables and quantifies how much information about one variable (e.g., LSK and/or miR-142 state) is contained in another one (e.g., expression level of individual genes). The higher the MI value, the higher is the amount of information. Thus, we computed the MI of individual genes in the EMHGS and in the "remaining" transcriptome to distinguish between miR-142⁻/⁻BCR-ABL and miR-142⁺/⁺BCR-ABL LSKs. We found that the EMHGS had higher MI density than the "remaining" transcriptome (p = 0.003; Fig. 3h). Of note, there was no statistically significant difference in the number of DEGs contained in the EMHGS vs in the "remaining" transcriptome (Fig. 3h). The EMHGS also had higher MI density than the remaining transcriptome to distinguish normal (i.e., lacking BCR-ABL) miR-142⁻/⁻ vs miR-142⁺/⁺ LSKs (p < 0.001; Fig. 3i). Thus, we concluded that expression changes of EMHGS as induced by the acquisition of miR-142 deficit, contained sufficient information to capture the state-transition of BCR/ABL LSKs from CP to BC, thereby supporting the biological relevance of the EMHGS changes to the disease evolution.

## Metabolomic profile of BC-LSCs indicated increased bioenergetic oxidative metabolism

To support functionally the relevance of EMHGS changes to BC transformation as identified by our state-transition approach, we then firstly conducted an unbiased metabolomic profiling of LSC-enriched cell populations from *miR-142⁻/⁻BCR-ABL* and *miR-142⁺/⁺BCR-ABL* mice, as previously reported[45,46]. To obtain enough cells, we combined BM LSK-enriched Lin⁻c-Kit⁺ cells harvested from 2–3 individual *miR-142⁻/⁻BCR-ABL* mice (n = 24) and from 2–3 individual *miR-142⁺/⁺BCR-ABL* (n = 18) mice at four weeks after BCR-ABL induction to obtain respectively 9 and 7 different "pooled" samples with similar cell numbers. The details on total and annotated metabolites detected in these samples are summarized in Supplementary Tables 2 and 3 and Supplementary Data 1 and 2. Comparing *miR-142⁻/⁻BCR-ABL* with *miR-142⁺/⁺BCR-ABL* cells, we identified 301 differentially abundant metabolites (i.e., acyl carnitines, fatty acids, fatty alcohols, lipids, amino acids and biogenic amines, nucleotides and derivatives, organic acids and sugars; p < 0.05;

Supplementary Data 2). Unsupervised clustering of the significantly different metabolites separated *miR-142⁻/⁻BCR-ABL* from *miR-142⁺/⁺BCR-ABL* cells (Fig. 4a).

Among the wealth of data obtained from metabolomic analysis and summarized in Supplementary Table 2 and 3 and Supplementary Data 1 and 2, we noticed a significant decrease of long chain complex lipids (triglycerides, phospholipids and sphingolipids), and a significant increase of single chain phospholipids (C14-C17), medium to long chain fatty acids (FAs), hydroxy FAs, fatty alcohol, ketone, amino FAs and a long chain acyl carnitine (3-hydroxypalmitoyl carnitine) in miR-142⁻/⁻BCR-ABL compared with miR-142⁺/⁺BCR-ABL cells (Fig. 4b). These data were suggestive of an increased lipid catabolism and FAO[37] in miR-142⁻/⁻BCR-ABL cells. Furthermore, in miR-142⁻/⁻BCR-ABL cells, we observed significant increase in metabolites of mitochondrial electron transport chain [i.e., NAD+ (p = 0.00025), the NAD+/NADH ratio (p = 0.018, Supplementary Table 3) and FAD (p = 0.049); Fig. 4c], suggesting enhanced OxPhos. Finally, increased ROS scavengers [e.g., carnosine (p = 0.00041) and β-alanine (p = 0.014)] and decreased oxidized glutathione in miR-142⁻/⁻BCR-ABL cells suggested that, despite the enhanced OxPhos, the oxidative stress was reduced[41].

In summary, the metabolomics analysis also indicated an increase in FAO and OxPhos (Fig. 4d), thereby supporting the RNA-seq results.

## miR-142 deficit increases FAO in murine and human BC LSCs

To support further the relevance of RNA-seq and metabolomic profiling results to both mouse and human disease, we then conducted parallel functional metabolic assays both in mouse (LSK) and human (CD34⁺CD38⁻) LSC-enriched cell populations harvested respectively from CP and BC CML mice and patients. To assess FAO levels, we measured the oxidation rate of ³H-palmitic acid analysis in *miR-142⁻/⁻BCR-ABL* vs *miR-142⁺/⁺BCR-ABL* LSKs and in human BC CD34⁺CD38⁻ vs CP CD34⁺CD38⁻ cells. Consistent with the metabolomic results, we observed approximately 2-fold increase in the FAO rate in *miR-142⁻/⁻BCR-ABL* LSKs and BC CD34⁺CD38⁻ cells compared respectively with *miR-142⁺/⁺BCR-ABL* LSKs and CP CD34⁺CD38⁻ cells (Fig. 5a). As FAO was found increased, we then assessed the levels of carnitine palmitoyltransferase I (Cpt1), which is a key FAO rate-limiting enzyme. The CPT1 protein family includes three isoforms: CPT1A, CPT1B, and CPT1C that are differentially expressed in various tissues. CPT1A and 1B are enriched in hematopoietic cells[47-51]. CPT1 is located in the outer mitochondrial membrane and catalyzes the transfer of the acyl group from a long-chain fatty acyl-CoA to l-carnitine to form palmitoylcarnitine and allows FAs to be transported into the mitochondria where they undergo β−oxidation[47]. MiR-142 reportedly targets CPT1A[13]. Consistent with this, CPT1A levels were found to be increased in *miR-142⁻/⁻BCR-ABL* LSKs and human BC CD34⁺CD38⁻ cells compared with their respective mouse (*miR-142⁺/⁺BCR-ABL* LSKs) and human (CD34⁺CD38⁻) CP controls (Fig. 5b). Surprisingly, however, we also noted

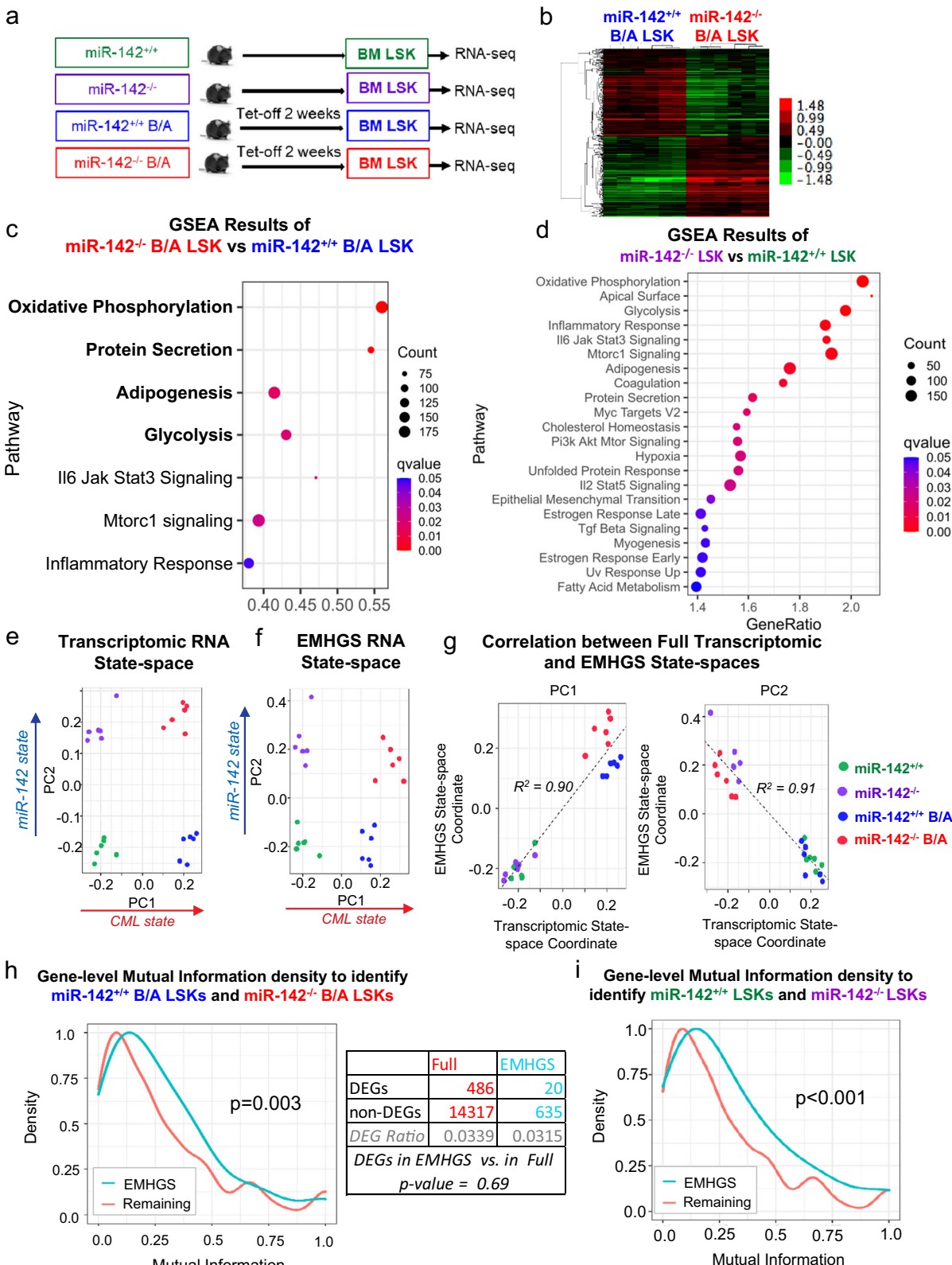

**a** Schematic of miR-142+/+, miR-142-/-, miR-142+/+ B/A, miR-142-/- B/A mice with Tet-off 2 weeks, BM LSK, RNA-seq.

**b** Heatmap of miR-142+/+ B/A LSK and miR-142-/- B/A LSK.

**c** GSEA Results of miR-142-/- B/A LSK vs miR-142+/+ B/A LSK

**d** GSEA Results of miR-142-/- LSK vs miR-142+/+ LSK

**e** Transcriptomic RNA State-space

**f** EMHGS RNA State-space

**g** Correlation between Full Transcriptomic and EMHGS State-spaces

**h** Gene-level Mutual Information density to identify miR-142+/+ B/A LSKs and miR-142-/- B/A LSKs. p=0.003

|            | Full   | EMHGS  |
|------------|--------|--------|
| DEGs       | 486    | 20     |
| non-DEGs   | 14317  | 635    |
| *DEG Ratio* | 0.0339 | 0.0315 |
| *DEGs in EMHGS vs. in Full p-value = 0.69* | | |

**i** Gene-level Mutual Information density to identify miR-142+/+ LSKs and miR-142-/- LSKs. p<0.001

upregulation of CPT1B (Fig. 5b), which is not a miR-142 target. To this end, CPT1 levels are reported to be lower in mice lacking the transcription factor Nrf2 (i.e., Nrf2-/-) compared with the Nrf2wt/wt mouse[52,53], implying a possible NRF2-dependent CPT1 transcription regulation. Accordingly, we observed increased NRF2 in *miR-142-/- BCR-ABL* LSKs and human BC CD34+CD38- cells compared with their respective controls (Fig. 5b; Supplementary Fig. 7a).

NRF2 is a protein that, once phosphorylated, translocates to the nucleus where it transactivates several target genes[54,55]. PKCα phosphorylates and stabilizes NRF2, thereby rescuing it from ubiquitination and degradation[56]. PKCα is a reported miR-142 target[57]. Accordingly, we observed higher levels of PKCα and in turn of NRF2 (Supplementary Fig. 7a), PKCα/NRF2 interaction (Supplementary Fig. 7b) and p-NRF2 (Supplementary Fig. 7a), reduced NRF2 ubiquitination (Supplementary

**Fig. 3 | State-transition of CP-LSCs to BC-LSCs is characterized by changes in the expression of genes involved in the bioenergetic oxidative metabolism.**
**a** Experimental design. BM LSKs from *miR-142⁺/⁺* (wt), *miR-142⁻/⁻* (miR-142 KO), *miR-142⁺/⁺BCR-ABL* (CP CML) and *miR-142⁻/⁻BCR-ABL* (BC CML) mice (2 weeks after Tet-off and BCR-ABL induction) were sorted for RNA-seq. **b** 506 differentially expressed genes, 211 upregulated and 295 downregulated, were identified in *miR-142⁻/⁻/BCR-ABL* LSKs vs *miR-142⁺/⁺/BCR-ABL* LSKs. **c, d** The upregulated gene sets by GSEA using Hallmark gene sets in **c** *miR-142⁻/⁻BCR-ABL* LSKs vs *miR-142⁺/⁺BCR-ABL* LSKs and **d** *miR-142⁻/⁻* LSKs vs *miR-142⁺/⁺* LSKs. For GSEA, see methods for details. **e** Using singular value decomposition (SVD) on the whole transcriptome, a state-space was constructed that separates the four experimental conditions. PC1 separated CML from non-CML samples, whereas PC2 separated miR-142 KO from wt samples. **f** By performing SVD on the 655 genes of the EMHGS, the EMHGS state-space was constructed and found to be highly similar to the whole transcriptome state-space. **g** Plotting the whole transcriptome and EMHGS state-space coordinates for both PC1 and PC2 confirmed that the state-spaces were similar ($R^2 = 0.90$ and $R^2 = 0.91$

respectively). **h** The density of mutual information (MI) of the EMHGS genes vs the remaining transcriptome (whole transcriptome minus EMHGS) was determined by calculating the MI for each gene using the *miR-142⁺/⁺BCR-ABL* LSK (CP-LSK) and *miR-142⁻/⁻BCR-ABL* LSK (BC-LSK) samples. The one-sided Wilcoxon rank sum test was used to determine that the EMHGS had higher MI density (MI distribution shifted toward higher values) compared to the remaining transcriptome ($p = 0.003$). The table compared the number of BC-LSK vs CP-LSK differentially expressed genes (DEGs) found in the "remaining" transcriptome (full transcriptome minus the EMHGS genes) with the number of DEGs found in the EMHGS. DEGs contained in the EMHGS showed no significant difference compared to the DEGs contained in the "remaining" transcriptome (hypergeometric p-value = 0.69). **i** The MI density of the EMHGS genes vs the remaining transcriptome (whole transcriptome minus EMHGS) was determined by calculating the MI for each gene using the *miR-142⁺/⁺* LSK (WT) and *miR-142⁻/⁻* LSK (KO) samples (one-sided Wilcoxon rank sum test, $p < 0.001$). For **e–i**, source data are provided as a Source Data file.

Fig. 7c), and increased both nuclear and cytoplasmic NRF2 (Supplementary Fig. 7d, e, left) in *miR-142⁻/⁻BCR-ABL* LSKs and human BC CD34⁺CD38⁻ cells compared with their respective mouse and human controls. Using a chromatin immunoprecipitation assay, we also showed that NRF2 was enriched on the *CPT1B* promoter and associated with higher *CPT1B* mRNA expression level both in the *miR-142⁻/⁻BCR-ABL* LSKs and human BC CD34⁺CD38⁻ cells as compared with their respective mouse and human CP CML controls (Fig. 5c; Supplementary Fig. 7f). Treatment with CpG-M-miR-142, a miR-142 mimic ODN (see methods for details) or with PIK75, a PKCα inhibitor, decreased NRF2 nuclear localization (Supplementary Fig. 7d, e, right), decreased the NRF2/PKCα interaction (Supplementary Fig. 7g) and increased NRF2 ubiquitination (Supplementary Fig. 7h), thereby resulting in decrease of NRF2, CPT1A and CPT1B levels (Supplementary Fig. 7i–k) and in turn of FAO (Supplementary Fig. 7l). Consistent with these results, treatment with the CPT1 inhibitor etomoxir (ETO, 5 μM) or CPT1A/B knockdown with CPT1A and CPT1B siRNA (20 nM) in mouse *miR-142⁻/⁻BCR-ABL* LSKs and human BC CD34⁺CD38⁻ cells significantly suppressed FAO and OxPhos levels (Supplementary Fig. 7m–o). These results supported that miR-142 deficit resulted in direct and indirect upregulation of the CPT1 isoforms and in turn increased FAO both in murine and human BC-LSCs compared with the respective CP-LSCs.

To determine how changes in FAO impacted on LSC activity, next, we treated *miR-142⁻/⁻BCR-ABL* LSKs (CPT1^high) with ETO for 72 h, to inhibit CPT1 activity[58] or with CTP1A/1B siRNAs to KD CPT1A/1B. Both treatments resulted in increased apoptosis, reduced cell growth and/or colony-forming cells (CFC) (Supplementary Fig. 8a–c) compared with their respective controls. Furthermore, we treated LSKs selected from *miR-142⁺/⁺BCR-ABL* or *miR-142⁻/⁻BCR-ABL* mice 3 weeks after BCR-ABL induction, with ETO (25 μM, 96 h)[58] or vehicle ex vivo, and transplanted these cells into congenic wt B6 mice via tail venous infusion (Supplementary Fig. 8d). Recipients of ETO-treated *miR-142⁻/⁻BCR-ABL* LSKs had reduced WBC counts and circulating engraftment rates at 4 weeks and survived significantly longer (median survival: 85 vs 51.5 days, $p = 0.0006$; Supplementary Fig. 8e, f) than recipients of vehicle-treated *miR-142⁻/⁻BCR-ABL* LSKs. Of note, no difference in survival was instead observed for the recipients of ETO-treated vs vehicle-treated *miR-142⁺/⁺BCR-ABL* LSKs (median survival: 214 vs 192 days, $p = 0.7$; Supplementary Fig. 8f), suggesting higher relevance of this metabolic process to the activity of BC-LSCs compared with that of CP-LSCs.

### miR-142 deficit increases mitochondrial fusion and OxPhos activity in murine and human BC LSCs
Next, we focused on OxPhos. Using Agilent Seahorse assays, we observed a significant increase in OxPhos [measured as oxygen consumption rate (OCR); Fig. 5d, left] and ATP levels (Supplementary Fig. 9a, left) but not in glycolysis [measured as extracellular

acidification rate (ECAR); Supplementary Fig. 8g] in *miR-142⁻/⁻BCR-ABL* LSKs compared with *miR-142⁺/⁺BCR-ABL* LSKs. These changes were rescued by treatment with CpG-M-miR-142 (OCR levels, Fig. 5d, right; ECAR levels, Supplementary Fig. 8h; ATP levels, Supplementary Fig. 9b, left). Similar results were obtained in primary CD34⁺CD38⁻ cells (low miR-142) from BC CML patients compared with CD34⁺CD38⁻ cells (high miR-142) from CP CML patients (Fig. 5e; Supplementary Figs. 8i, j and 9a, b, right). Of note, despite the increase in OxPhos, levels of ROS, by-products of OxPhos that reportedly have the ability to damage LSC homeostasis and activity[59], were not increased. To explain this observation, we showed that once phosphorylated by PKCα, NRF2 translocates to the nucleus, transactivates ROS scavenger coding genes (e.g., HO-1 and NQO-1)[54,55], and reduces ROS both in mouse *miR-142⁻/⁻BCR-ABL* LSKs and human BC CML CD34⁺CD38⁻ cells compared respectively with *miR-142⁺/⁺BCR-ABL* LSKs and CP CML CD34⁺CD38⁻ cells (Supplementary Fig. 9c). Treatment with CpG-M-miR-142 or with PIK75 decreased HO-1 and NQO-1 expression and increased ROS levels (Supplementary Fig. 9d, e). In contrast, blocking of ROS induction by co-treatment with a ROS scavenger *N*-acetyl-l-cysteine reversed these effects (Supplementary Fig. 9f).

The increase in OxPhos without a consequent increase in ROS led us to postulate that changes in mitochondrial dynamics had occurred[60–62]. Mitochondria undergo fusion to meet the cellular bioenergetic needs and optimize OxPhos efficiency[63]. Mitofusin-1 (MFN1) is a key protein involved in mitochondrial fusion and a reported target of miR-142[63]. To this end, miR-142 deficit was associated with MFN1 upregulation and mitochondrial fusion in mouse *miR-142⁻/⁻BCR-ABL* LSKs and human BC CML CD34⁺CD38⁻ cells compared respectively with *miR-142⁺/⁺BCR-ABL* LSKs and CP CML CD34⁺CD38⁻ cells (Fig. 5f, g; Supplementary Fig. 9g, h). These changes were rescued by CpG-M-miR-142 (Fig. 5h, i; Supplementary Fig. 9i, j), which enhanced mitochondrial fission (i.e., fragmentation) and membrane polarization (Supplementary Fig. 9k, l). Using immunolabeled electron microscopy (EM) imaging, we showed that CpG-M-miR-142 also reduced the mitochondria-anchored MFN1 in both mouse *miR-142⁻/⁻BCR-ABL* LSKs and human BC CML CD34⁺CD38⁻ cells (Fig. 5j, k; Supplementary Fig. 9m). Therefore, taken altogether these results support that miR-142 deficit increased mitochondrial fusion, which associated with enhanced OxPhos, without increase in ROS production, in BC-LSCs as compared with CP-LSCs.

### In vivo delivery of miR-142 mimic reduced BC CML burden in GEMMs and PDXs
To correct miR-142 deficit in BC mice, we synthesized a miR-142 mimic ODN, hereafter called CpG-M-miR-142 (for full description see methods). Using a Cy3-labeled CpG-M-miR-142, we showed a valid uptake by ex vivo treated LSKs (Fig. 6a). We also confirmed in vivo uptake, by measuring Cy3-labeled CpG-M-miR-142 in LSKs harvested from normal

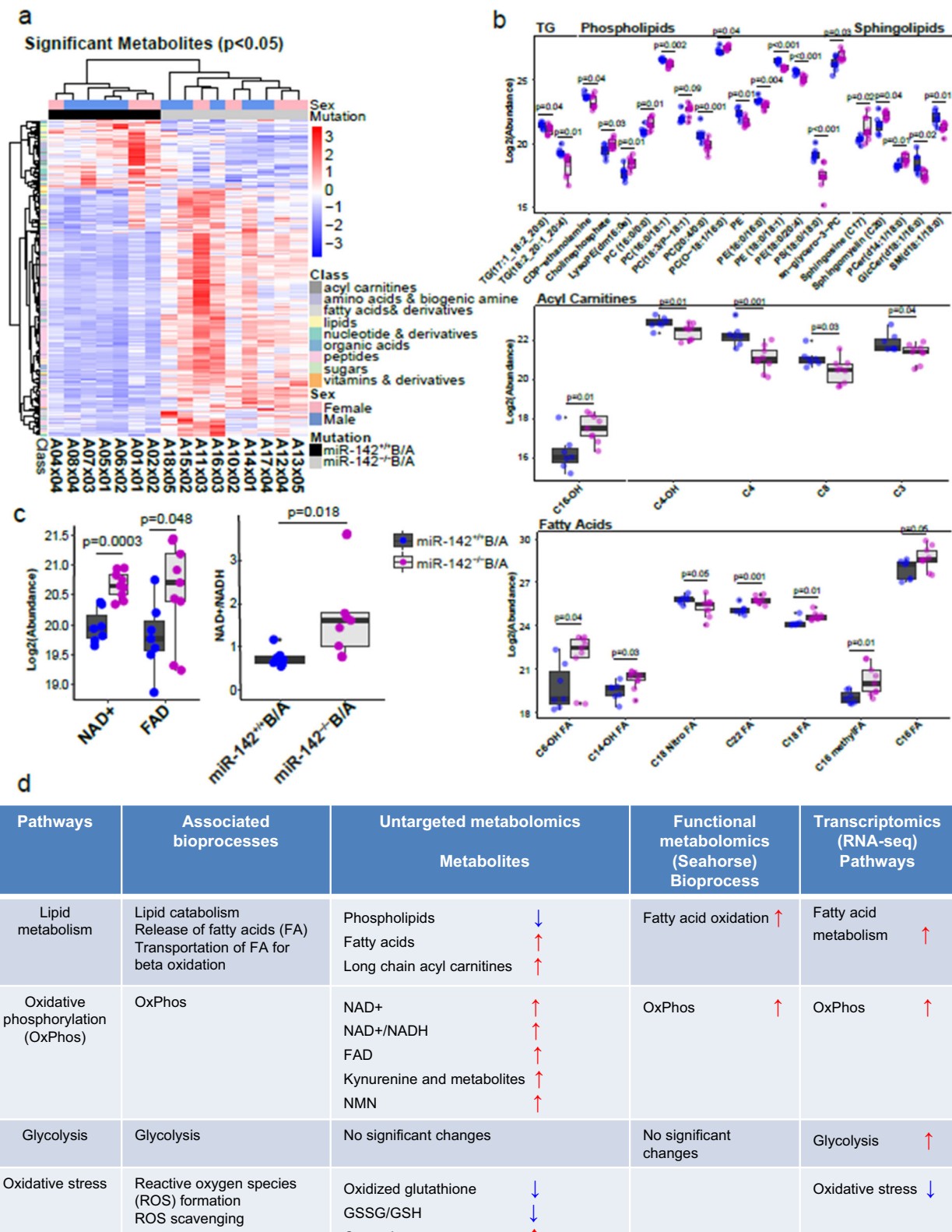

**d**

| Pathways | Associated bioprocesses | Untargeted metabolomics<br>Metabolites | | Functional metabolomics (Seahorse)<br>Bioprocess | | Transcriptomics (RNA-seq)<br>Pathways | |
|---|---|---|---|---|---|---|---|
| Lipid metabolism | Lipid catabolism<br>Release of fatty acids (FA)<br>Transportation of FA for beta oxidation | Phospholipids<br>Fatty acids<br>Long chain acyl carnitines | ↓<br>↑<br>↑ | Fatty acid oxidation | ↑ | Fatty acid metabolism | ↑ |
| Oxidative phosphorylation (OxPhos) | OxPhos | NAD+<br>NAD+/NADH<br>FAD<br>Kynurenine and metabolites<br>NMN | ↑<br>↑<br>↑<br>↑<br>↑ | OxPhos | ↑ | OxPhos | ↑ |
| Glycolysis | Glycolysis | No significant changes | | No significant changes | | Glycolysis | ↑ |
| Oxidative stress | Reactive oxygen species (ROS) formation<br>ROS scavenging | Oxidized glutathione<br>GSSG/GSH<br>Carnosine | ↓<br>↓<br>↑ | | | Oxidative stress | ↓ |

mice treated with a single dose (20 mg/kg) (Fig. 6b), or by measuring levels of miR-142 and targets in BM MNCs from *miR-142⁻/⁻ BCR-ABL* mice treated with CpG-M-miR-142 or CpG-scramble miRNA ODN control (SCR, 20 mg/kg/day, IV) for 3 weeks (Fig. 6c). The doses and administration schedule of the CpG-M-miR-142 were derived from our previous experience with similar CpG-miRNA compounds[28,64]. With a single dose of treatment, approximately 45% of the LSKs harvested

from the treated mice contained Cy3-CpG-M-miR-142 with approximately 50% decrease of the miR-142 target Cpt1a, compared with SCR (Fig. 6b). Upon treatment completion of 3 weeks, we observed a three-fold increase in expression of miR-142 in the BM MNCs from CpG-M-miR-142-treated mice compared to those from SCR-treated mice (Fig. 6c). Levels of miR-142 in *miR-142⁻/⁻ BCR-ABL* CML mice treated with CpG-M-miR-142 reached approximately 65% of the physiologic BM

**Fig. 4 | Metabolomic differences between CP-LSCs and BC-LSCs. a** Hierarchical unsupervised clustering represents 301 differently abundant (p < 0.05) metabolites (acyl carnitines, fatty acids, fatty alcohols, lipids, amino acids and biogenic amines, nucleotides and derivatives, organic acids and sugars) in BM Lin-c-Kit+ cells isolated from *miR-142⁻/⁻BCR-ABL* and *miR-142⁺/⁺BCR-ABL* mice (BCR-ABL were induced for 4 weeks by tetracycline withdrawal). Comparison between groups was performed by two-tailed, paired Student's t-test. **b** Box and Whisker plots depicting relative log2 abundances of significantly different complex lipids including triglycerides (TG), phospholipids and sphingolipids, fatty acyl carnitines and free fatty acids in *miR-142⁻/⁻BCR-ABL* (n = 9 biologically independent samples) vs *miR-142⁺/⁺BCR-ABL* (n = 7 biologically independent samples) Lin-c-Kit+ cells. Individual p-values for all 33 metabolites are available in Supplementary Data 2. **c** Box and Whisker plots showing the log2 relative abundances of NAD+ (complex I product) (p = 0.00025), FAD (complex II product, p = 0.048), and the NAD+/NADH ratio (Complex I product/substrate ratio, p = 0.018) in *miR-142⁻/⁻BCR-ABL* (n = 9 biologically

independent samples) vs *miR-142⁺/⁺BCR-ABL* (n = 7 biologically independent samples) Lin-c-Kit+ cells. For **b**, **c**, a two-sided Student's t-test was used to compare KO versus WT cells. Data presented in boxplots show the median as represented by the center line, with the range of the box from the 25th percentile to the 75th percentile, and whisker lines stretching to 1.5× the IQR below the 25th percentile and 1.5× the IQR above the 75th percentile. Any points in the boxplot not represented within these ranges (above 1.5× the 75th percentile or below 1.5× the 25th percentile) are represented as individual black points. All individual data points are overlayed on the boxplots for miR-142+/+ (blue) and miR-142−/− (purple) cells. For **b**, **c**, source data are provided as a Source Data file. **d** Table describing significant changes observed in metabolic pathways and biochemical processes upon comparing *miR-142⁻/⁻BCR-ABL* with *miR-142⁺/⁺BCR-ABL* cells using untargeted metabolomic analysis (Lin-c-Kit+ cells), functional seahorse analysis (LSK), and transcriptomic analysis (LSK).

levels measured in *miR-142⁺/⁺BCR-ABL* mice, with ~60% reduction in the miR-142 target Cpt1a (Fig. 6c). Notably, no hematologic or non-hematologic toxicities were observed in CpG-M-miR-142-treated mice during the following 8 weeks of monitoring.

To determine the therapeutic impact of rescuing of miR-142 deficit on delaying the BC-like transformation, next we treated a cohort of *miR-142⁻/⁻BCR-ABL* mice with CpG-M-miR-142 (20 mg/kg/day, iv) or SCR for 4 weeks, starting on the day after BCR-ABL induction (Fig. 6d). At the completion of treatment, compared with SCR-treated controls, we observed a significant decrease of BM blasts (p = 0.01; Fig. 6e, left) in the CpG-M-miR-142-treated mice, which also survived longer than the SCR-treated controls (median survival: 69 vs 25 days, p = 0.004; Fig. 6e, right). To assess the impact of CpG-M-miR-142 treatment on the LSC burden, we then transplanted 10⁶ BM cells from CpG-M-miR-142- or SCR-treated donors (CD45.2) into 2nd recipients (CD45.1). The recipients of BM from CpG-M-miR-142 treated donors had a significantly lower engraftment rate (CD45.2⁺: 63% vs 84%, p = 0.008) and longer survival (median survival: 68 vs 48 days, p = 0.002; Fig. 6f) than the recipients of BM from SCR treated donors.

Next, to validate how these results related to the human disease, we generated a patient-derived xenograft (PDX) by transplanting 2 × 10⁶ CD34⁺ cells from BC CML patients into NSG mice. Upon detecting engraftment (>5% blood human CD45⁺ cells), we treated these animals with CpG-M-miR-142 (20 mg/kg/day, iv) or SCR for 3 weeks (Fig. 6g). The CpG-M-miR-142-treated mice survived significantly longer than the SCR-treated mice (median survival: 62 vs 52 days, p = 0.006; Fig. 6h). BM cells from these treated mice were then transplanted into 2ⁿᵈ NSG recipient mice for post-treatment evaluation of residual LSC burden. Recipients of BM MNCs from CpG-M-miR-142-treated donors showed significantly lower engraftment rates and prolonged survival compared with recipients of BM MNCs from SCR-treated donors (median survival: not reached vs 71 days after 80 days post transplantation, p < 0.0001; Fig. 6i).

**CpG-M-miR-142 sensitized murine and human BC-LSCs to TKIs**

To assess the therapeutic activity of CpG-M-miR-142 in combination with TKI in vivo, we generated a disease-synchronized cohort of BC CML mice by transplanting 10⁶ BM MNCs from diseased *miR-142⁻/⁻BCR-ABL* mice (CD45.2) into congenic wt recipient mice (CD45.1). At 2 weeks post transplantation, these mice were treated with CpG-M-miR-142 (20 mg/kg/day, IV), nilotinib (NIL; 50 mg/kg/day, gavage), CpG-M-miR-142+NIL, or SCR + NIL for 3 weeks (Fig. 7a). We observed reduced leukemic burden and prolonged survival in CpG-M-miR-142 + NIL-treated mice compared with SCR + NIL-treated controls (median survival: unreached vs 130 days after monitoring for 150 days, p = 0.01; Fig. 7b). Of note, CpG-M-miR-142-treated mice also survived longer than SCR-treated mice (median survival: 107 vs 54 days, p = 0.026; Fig. 7b). Furthermore, 2nd recipients of BM MNCs from CpG-M-miR-142 + NIL-treated donors had a significantly longer survival

than recipients of BM MNCs from SCR + NIL-treated donors (median survival: unreached vs 110 days after monitoring for 180 days, p = 0.01; Fig. 7c).

We then transplanted 2 × 10⁶ CD34⁺ cells from BC CML patients into NSG mice. Upon detecting engraftment (>5% circulating human CD45⁺ cells), we treated the PDX mice with CpG-M-miR-142 (20 mg/kg/day, iv) + NIL (50 mg/kg/day, oral gavage) or SCR + NIL for 3 weeks (Fig. 7d). The CpG-M-miR-142 + NIL-treated mice survived significantly longer than SCR + NIL-treated mice (median survival: 47 vs 36 days, p = 0.008; Fig. 7e). Furthermore, 2nd recipients of BM MNCs from CpG-M-miR-142 + NIL-treated donors had significantly reduced engraftment (p < 0.0001) and prolonged survival (median survival: unreached vs 63.5 days after monitoring for 80 days, p < 0.0001; Fig. 7f) compared to recipients of BM MNCs from SCR + NIL-treated donors.

## Discussion

Understanding the mechanisms of BC transformation in CML may provide insight into LSC biology and evolution and may lead to the identification of a druggable "Achille's heel" in these otherwise highly treatment-resistant cells. From the observation that miR-142 levels are significantly lower in BC CML patients compared with CP CML patients, we postulated that an acquired miR-142 deficit played a role in BC transformation. To prove this hypothesis, we knocked out miR-142 in a CP CML mouse model. The *miR-142⁻/⁻BCR-ABL* mice developed a BC-like aggressive phenotype with 100% penetrance and a significantly shorter survival compared with the *miR-142⁺/⁺BCR-ABL* controls, that instead remained in CP CML for their lifespans. Importantly, LSCs from *miR-142⁻/⁻BCR-ABL* mice recapitulated the BC phenotypes when transplanted into 2nd wt recipient mice. Of note, whole genome sequencing of the *miR-142⁻/⁻BCR-ABL* LSKs failed to show any additional genomic "hits" when compared with that of *miR-142⁺/⁺BCR-ABL* LSKs, suggesting that miR-142 deficit alone sufficed in stably transforming CP-LSCs into BC-LSCs. The transforming role of miR-142 deficit in BCR-ABL expressing cells was further corroborated by the upregulation of Msi2, a predicted target of miR-142 that has been previously reported to contribute to BC transformation[32–34].

To gain insights into previously unveiled molecular mechanisms of BC-transformation in miR-142 deficient CML mice, we elected to conduct "bulk" and sc- RNA-seq analyses on *miR-142⁻/⁻BCR-ABL* LSKs vs *miR-142⁺/⁺BCR-ABL* LSKs. These analyses showed a change in the BM LSK landscape which became skewed toward myeloid differentiation with loss of T cell differentiation. Importantly, we noted a significant enrichment of genes involved in oxidative metabolic processes (e.g., FAO and OxPhos), which are reportedly critical for LSC homeostasis and activity. Using elements of state transition and MI theories, we showed that the metabolism gene sets that were upregulated in *miR-142⁻/⁻BCR-ABL* LSKs compared with *miR-142⁺/⁺BCR-ABL* LSKs contained higher per-gene information than the remaining transcriptome to

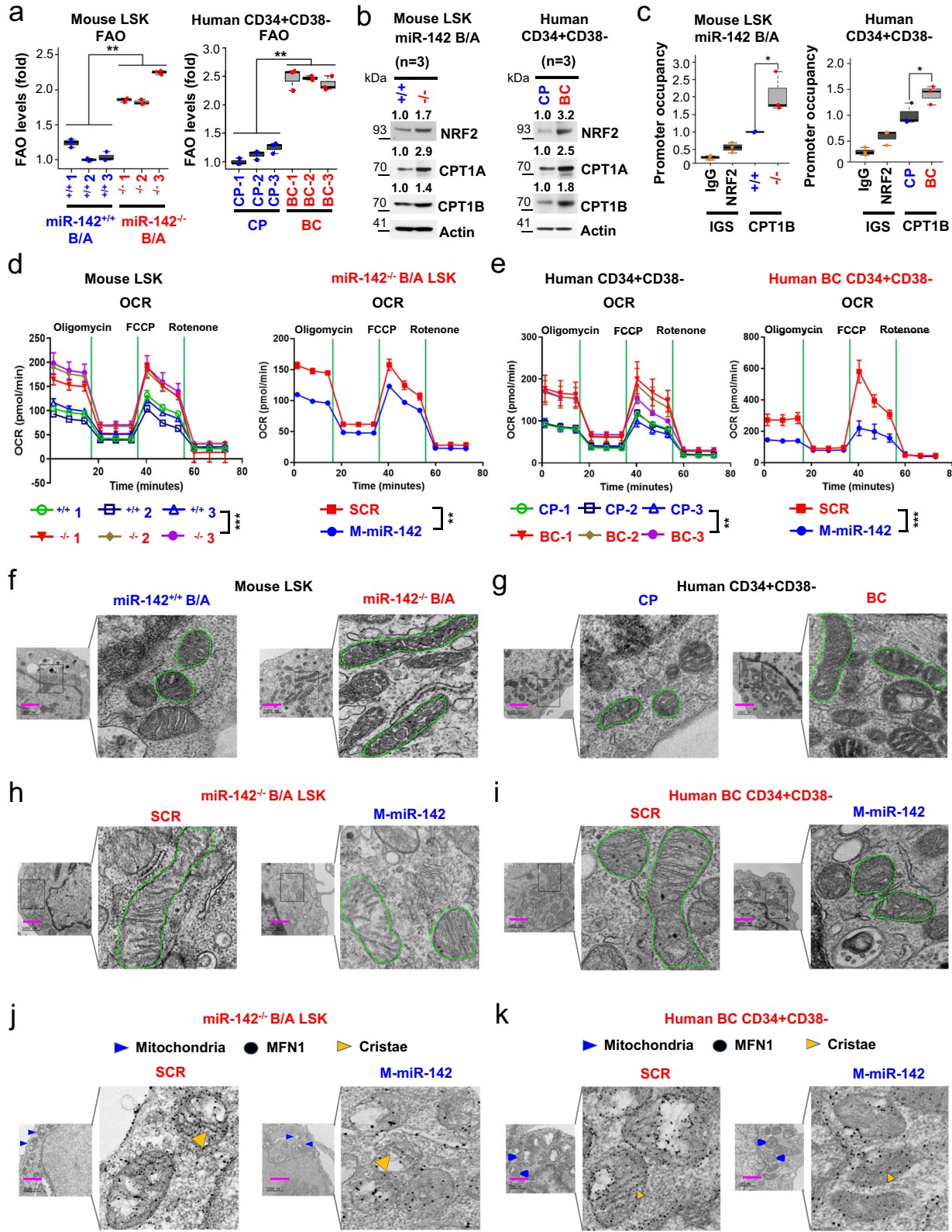

distinguish BC from CP LSKs, supporting the biological relevance of these genes to BC transformation. To confirm these results, we then pursued both unbiased metabolomic profiling and functional metabolic assays using BC vs CP LSC-enriched cell populations and obtained evidence supporting a significant increase in FAO and OxPhos in BC-LSCs compared with CP-LSCs induced by the acquisition of the miR-142 deficit.

From a mechanistic standpoint, we demonstrated that miR-142 deficit enhanced FAO via upregulation of both Cpt1a and Cpt1b, two FAO rate-limiting step enzymes[47–51]. CPT1A is a reported target of miR-142[13], hence the increase of this protein in *miR-142⁻/⁻ BCR-ABL* LSKs was not surprising. CPT1B instead is not a miR-142 target, and therefore its upregulation in *miR-142⁻/⁻ BCR-ABL* LSKs was somewhat unexpected. Nevertheless, we demonstrated that the miR-142 target PKCα was

**Fig. 5 | Effects of miR-142 expression on mitochondrial morphology and metabolism in leukemic stem cells. a–c** LSKs and CD34⁺CD38⁻ cells were respectively isolated from *miR-142⁺/⁺BCR-ABL* vs *miR-142⁻/⁻BCR-ABL* mice (**left**) or from human CP vs BC CML patients (**right**). **a** Levels of FAO (left, p = 0.0048; right, p = 0.0023). **b** Immunoblotting of NRF2, CPT1A and CPT1B protein. Three samples of each group were pooled for the assay (n = 3 biologically independent samples). Densitometry quantifications (fold) are shown on the top. Source data are provided as a Source Data file. **c** ChIP assay to measure NRF2 binding to CPT1B promoter levels (left, p = 0.016; right, p = 0.0307). Box plot with median value and first/third quartiles and whiskers together with minimum and maximum values are shown (n = 3 biologically independent samples). **d, e** Levels of OxPhos (as indicated by OCR levels in Seahorse assay) in **d** LSKs from *miR-142⁺/⁺BCR-ABL* vs *miR-142⁻/⁻BCR-ABL* mice (**d**, left) or **e** CD34⁺CD38⁻ cells from human CP vs BC CML patients (**e**, left). Mouse *miR-142⁻/⁻BCR-ABL* LSKs (**d**, right, n = 3 biologically independent samples) or human CD34⁺CD38⁻ BC CML cells (**e**, right, n = 3 biologically independent samples) were treated with SCR control or CpG-M-miR-142 (2 μM) for 24 h (**d**: left, p = 0.0008; right, p = 0.0014; **e**: left, p = 0.0054; right, p = 0.0009). For **a–e**, results shown represent

from one of the three independent experiments are shown (n = 3). Comparison between groups was performed by one-tailed, unpaired t-test. Results shown represent mean ± SD. Significance values: *p < 0.05; **p < 0.01; ***p < 0.001; ****p < 0.0001; ns, not significant. Effects of miR-142 expression on mitochondria fusion in LSCs. Evidence of mitochondria fusion in **f** LSKs from *miR-142⁺/⁺BCR-ABL* vs *miR-142⁻/⁻BCR-ABL* mice and **g** CD34⁺CD38⁻ cells from human CP vs BC CML patients. **h–k** Effects of correction of miR-142 deficit on mitochondria fusion. Indicated cells were treated with SCR control or CpG-M-miR-142 (500 nm) for 24 h. **h, i** Mitochondria morphology was shown by electron microscope. **j, k** Levels of MFN1 protein expression (black dots) inside mitochondria and effect of CpG-M-miR-142 visualized by immunolabeling-electron microscope. For **a–e**, source data are provided as a Source Data file. For **f–k**, scale bar, 1000 nm, results from one of the three independent experiments are shown (n = 3). miR-142⁺/⁺B/A *miR-142⁺/⁺BCR-ABL*, miR-142⁻/⁻B/A *miR-142⁻/⁻BCR-ABL*, CP chronic phase, BC blast crisis, FAO fatty acid oxidation, ChIP chromatin immunoprecipitation, OCR oxidative consumption rate, M-miR-142 CpG-M-miR-142, SD standard deviation.

increased in *miR-142⁻/⁻BCR-ABL* LSKs. This protein stabilizes and increases Nrf2, that once translocated into the nucleus, transactivates the *Cpt1b* gene. Of note, these changes were rescued by restoring endogenous levels of miR-142 with a newly synthesized miR-142 mimic ODN (i.e., CpG-M-miR-142).

We also demonstrated that OxPhos was significantly enhanced in *miR-142⁻/⁻BCR-ABL* LSKs through an increase in mitochondrial fusion induced by upregulation of the miR-142 target Mfn1. This is a GTPase protein that participates in mitochondrial fusion and contributes to the formation of structural and functional mitochondrial networks, resulting in higher OxPhos levels and efficiency. Utilizing EM imaging and other functional assays, we showed that higher levels of Mpn1 was associated with mitochondrial fusion in both murine (i.e., *miR-142⁻/⁻BCR-ABL* LSKs) and human (i.e., CD34⁺CD38⁻ blasts) BC-LSCs compared with CP-LSCs. Importantly, as also suggested by the metabolomic analysis, the increase in OxPhos in BC-LSCs was not associated with a significant production of ROS, which reportedly damage LSC homeostasis and activity[59]. Of note, these changes were rescued by treatment with CpG-M-miR-142, which downregulated Mpn1, induced mitochondrial fission (i.e., fragmentation), decreased OxPhos, enhanced ROS production and in turn increased the apoptotic rate of BC-LSCs.

Taken altogether, our results support that an acquired miR-142 deficit alone induced BC transformation in LSCs via multifaceted and interconnected mechanisms that increase FAO and mitochondrial fusion and that culminate in enhanced OxPhos, with a minimized ROS production (see graphical abstract). These results also suggested that mitochondrial fusion as observed both in human and mouse BC cells was associated with increased metabolic functions (i.e., FAO and OxPhos) as also reported by others[61,62]. Of note, the role of miR-142 as a modifier of bioenergetic oxidative metabolism was previously described in immune cells. To this end, Sun et al., showed that miR-142 plays a key role in controlling metabolic reprogramming, i.e., the OxPhos-to-glycolysis switch, during dendritic cell activation and immunogenic response[13]. Interestingly, our results are also reminiscent of findings recently reported by Bonnay et al., showing that increased OxPhos and mitochondrial fusion induced immortalization and transformation of neural stem cells in a Drosophila model[65]. Thus, the aberrant reprogramming of oxidative metabolism may be a potentially conserved mechanism of cell transformation across different tissues and species.

Thus, given the "key" role of miR-142 deficit in BC transformation, we then asked if this deficit itself was a druggable BC-LSCs' "Achille's heel". To this end, we designed and optimized a novel mimic ODN, called CpG-M-miR-142. Of note, RNA therapeutics have previously been considered with skepticism due to several anticipated drawbacks including rapid RNase degradation, poor delivery of negatively charged RNA across the hydrophobic cytoplasmic membrane, and strong immunogenicity of exogenous RNA[66]. Nevertheless RNA-based

drugs, including non-coding RNAs, have the advantage of aiming at targets that are often "undruggable" with small molecules. Furthermore, important progresses made in this area have led to FDA approvals of mRNA vaccines and other RNA-based therapeutics[67].

Of note, while delivery of RNA therapeutics is often achieved with nanoparticles, for the in vivo targeting of miR-142 deficit, we elected to use a different approach. Leveraging a platform previously reported by our group, we designed a 2′-O-methyl-RNA modified moiety formulated into a cDNA phosphorothioated CpG-backbone[28,68,69]. One of the advantages of using CpG ODN as a delivery system for RNA therapeutics over nanoparticle formulations is a potentially more selective uptake of the ODNs by the targeted organs (i.e., TLR-9 expressing BM and spleen hematopoietic cells), whereas nanoparticle-encapsulated drugs often undergo a non-specific hepatic first-pass. To this end, we showed that our CpG-M-miR-142 drug was efficiently taken up by both murine and human LSCs in vitro and in vivo, resulting in meaningful pharmacological activity (i.e., increase of miR-142 and decrease of miR-142 target levels). Furthermore, CpG-M-miR-142 treatment alone and in combination with TKIs significantly increased miR-142 activity, decreased disease growth, and improved survival in both BC murine and PDX models, in the absence of toxicity. Importantly, CpG-M-miR-142 significantly decreased LSC burden as demonstrated with 2nd transplant experiments. Combination of CpG-M-miR-142 and TKI resulted in a significantly longer survival of 2nd recipient mice, with 100% of the BM recipients from PDX donors treated with the combination being alive at approximately 3 months post transplantation compared with 0% of the BM recipients from PDX donors treated with SCR + TKI. Of note, treatment with CpG-M-miR-142 was administered only for 4 weeks, since daily retroorbital injections of the drug prevented a longer administration schedule. Once the treatment was discontinued, mice eventually relapsed with BC. Nevertheless, despite this relatively narrow schedule, in secondary transplant experiments, we were able to show that CpG-M-miR-142 decreased LSC burden, as supported by the longer survival of the recipients of BM from miR-142 mimic-treated donors compared with the recipients of BM from SCR-treated donors. Thus, it is likely that a longer CpG-M-miR-142 administration alone or in combination with TKIs may be of further clinical benefit and may lead to complete eradication of LSCs and achievement of a cure.

Taken altogether, these results support that miR-142 deficit alone not only promotes BC transformation via metabolic reprogramming of LSCs but may also represent a potentially novel therapeutic target to eradicate BC-transformed CML.

## Methods
### Human samples
CP and BC CML samples were obtained from patients who had not received TKI treatment at the City of Hope National Medical Center

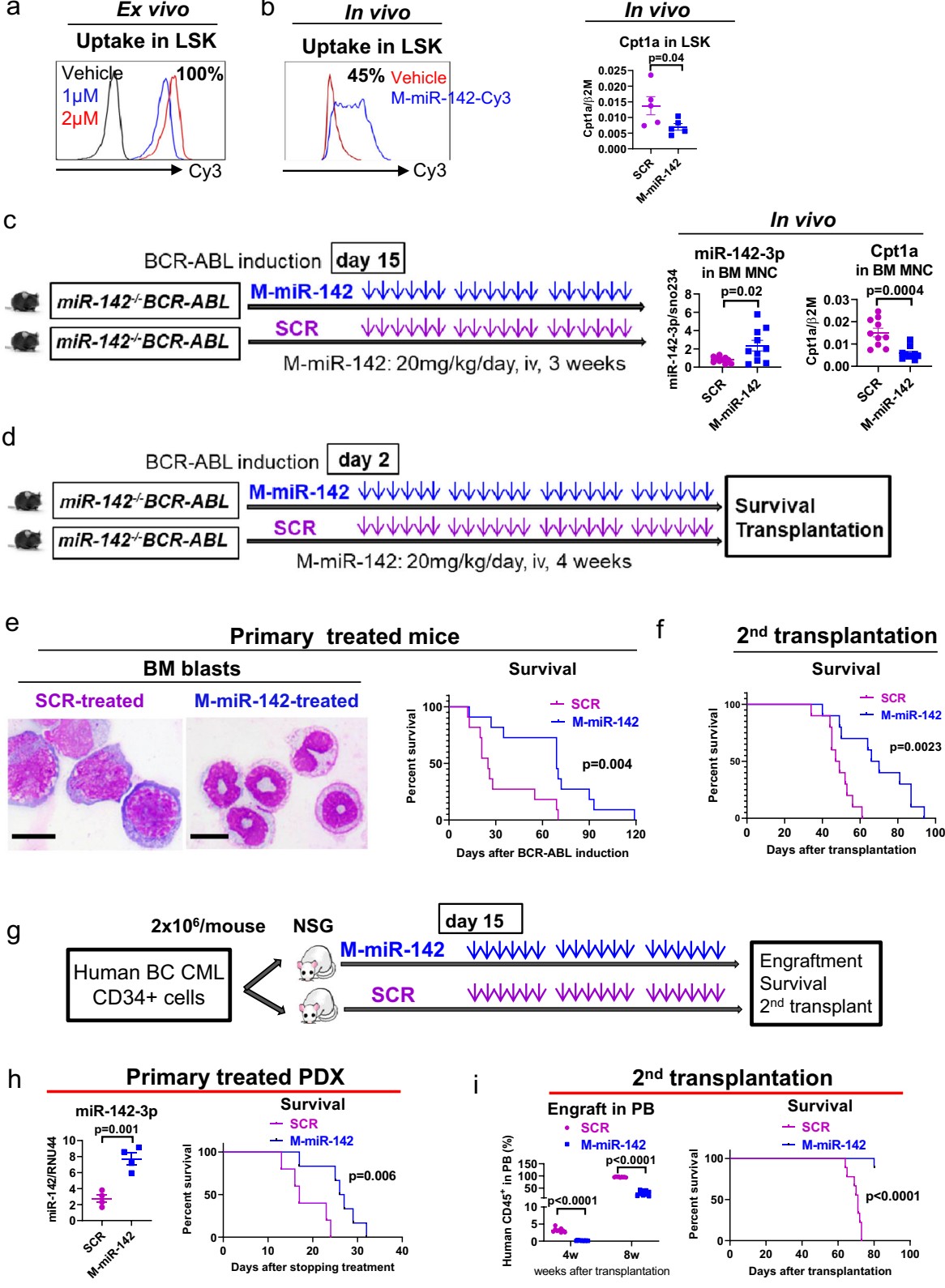

(COHNMC). All CML samples used in this study are P210 BCR−ABL positive, as confirmed by FISH analysis and qPCR. Mononuclear cells (MNCs) were isolated using Ficoll separation. All CML patients and healthy donors signed an informed consent form. Sample acquisition was approved by the Institutional Review Board (IRB, #18067 protocol) at the COHNMC, in accordance with an assurance filed with and approved by the Department of Health and Human Services and met all requirements of the Declaration of Helsinki.

## Animal studies

SCLtTA/BCR-ABL transgenic mice [CD45.2 C57BL/6(B6), hereafter called BCR-ABL][26,27] were maintained on tetracycline (tet)-containing water at 0.5 g/liter. Withdrawal of tet results in expression of BCR−ABL and generation of a CP CML-like disease in these mice[26,27]. MiR-142 KO (i.e., *miR-142*[−/−], CD45.2 B6) mice were generated by our collaborator Dr. Mark Boldin[12]. *MiR-142*[+/−]*BCR-ABL* and *miR-142*[−/−]*BCR-ABL* (CD45.2 B6) mice were generated by crossing *miR-142*[−/−]

**Fig. 6 | In vivo delivery of miR-142 mimic reduced BC CML burden in GEMMs and PDXs.** In vivo **a** Ex vivo uptake of Cy3-labeled CpG-M-miR-142 in LSK cells by flow cytometry. **b** In vivo uptake of Cy3-labeled CpG-M-miR-142 by flow cytometry and mRNA expression of the miR-142 target Cpt1a by Q-RT-PCR in LSKs from normal mice treated with a single dose of Cy3-labeled CpG-M-miR-142 (20 mg/kg) (n = 5 biologically independent samples per group). **c** MiR-142$^{-/-}$BCR-ABL CML mice (BCR-ABL were induced for two weeks by tet-off) were treated with CpG-M-miR-142 or SCR (20 mg/kg/day, IV) for 3 weeks, then levels of miR-142 and its target Cpt1a in BM MNCs from these treated mice were measured by Q-RT-PCR (n = 10 biologically independent samples per group). **d** Experimental design. To determine the impact of in vivo rescuing of miR-142 deficit on the BC transformation rate, a cohort of miR-142$^{-/-}$BCR-ABL mice were treated with CpG-M-miR-142 (20 mg/kg/day, iv) or SCR for 4 weeks, starting on the day after tet-off induced BCR-ABL induction. **e** At treatment completion, white blood cell (WBC) counts, circulating LSKs by flow cytometry, BM blasts by microscopy (scale bar: 10 μM), and survival of the CpG-M-miR-142-treated vs SCR-treated mice are shown. **f** To assess the impact of CpG-M-miR-142 treatment

on the LSC burden, 10$^6$ BM cells from the above CpG-M-miR-142- or SCR-treated mice (CD45.2) were transplanted into secondary (2nd) recipient mice (CD45.1). WBC counts, engraftment (CD45.2$^+$) rates, and survival of the 2nd recipients are shown. **g** Experimental design. BC CML patient-derived xenograft (PDX) mice were generated by transplanting 2×10$^6$ CD34$^+$ cells from BC CML patients into NSG mice. Upon detecting engraftment (>5% blood human CD45$^+$ cells) at two weeks post transplantation, these mice were treated with CpG-M-miR-142 (20 mg/kg/day, iv) or SCR for 3 weeks. **h** BM miR-142 levels (n = 4 biologically independent samples per group) and survival of these treated mice (n = 6 mice per group) are shown. **i** BM cells from these treated mice were transplanted into 2nd NSG recipient mice (10$^6$/mouse) and PB engraftment (human CD45$^+$) rates and survival of the 2nd recipient mice (n = 9 mice per group) are shown. For **b**, **c**, **e**, **f** and **h**, **i**, source data are provided as a Source Data file. For **a**, **b**, **e**, results from one of the three independent experiments are shown (n = 3). Comparison between groups was performed by two-tailed, unpaired t-test. Survival curve was compared by Log-rank test. Results shown represent mean ± SEM.

with BCR-ABL mice. The genotyping of the above mice was performed by Transnetyx. To evaluate the impact of upregulating miR-142 by our homemade drug (i.e., CpG-M-miR-142) on leukemia progression and LSC burden, a cohort of BC CML mice with a similar leukemia onset time were generated by transplanting BM cells (from both tibias and femurs) from the diseased miR-142$^{-/-}$BCR-ABL mice (CD45.2 B6, BCR-ABL expression was induced for two to four weeks by tet withdrawal as indicated) into congenic recipient mice (CD45.1 B6). Eight-week-old CD45.1 B6 (from Charles River) mice were irradiated at 6 Gy within 24 h before transplantation and used as recipients to allow tracking of CD45.2 B6 donor cells. To study the effect of CpG-M-miR-142 on human BC CML cells, a patient-derived xenograft (PDX) model was generated by transplanting human BC CML CD34+ cells into NSG (2 Gy, The Jackson Laboratory) mice. The number of mice for each study group was chosen based on the expected endpoint variation (i.e., engraftment rate and latency period of leukemia) and on the availability of mice from different strains. To achieve a meaningful comparison, we matched miR-142$^{-/-}$BCR-ABL and miR-142$^{+/+}$BCR-ABL mice not only for age and gender, but also for SCL and BCR-ABL level (by Q-RT-PCR). Investigators were blinded to mouse genotype while performing treatment or monitoring for engraftment or survival. All the experimental mice were housed in 68−79 F temperature and 30-70% humidity, in a 12:12-h light:dark cycle. Mice are group-housed in individually ventilated cages (Optimice, Animal Care Systems, Centennial, CO). Mice are allowed free access to rodent chow (no. 5053, LabDiet, St Louis, MO), and reverse-osmosis−purified water. Mouse care and experimental procedures were performed in accordance with federal guidelines and protocols and were approved by the Institutional Animal Care and Use Committee at City of Hope.

### Flow cytometry analyses
Human CD34$^+$ cells were selected using the indirect CD34 microbead kit (Miltenyi Biotec, San Diego, CA) and CD34$^+$CD38$^+$ committed progenitors and CD34$^+$CD38$^-$ primitive progenitors were obtained by flow cytometry sorting after staining with antibodies against human CD34 and CD38 (Supplementary Table 4) according to the manufacturer's protocol. To determine engraftment rate of human CML cells in NSG mice, PB, BM and spleen cells were stained with anti-human CD45, CD33 and CD34 antibodies (Supplementary Table 4). Mouse cells were obtained from PB, BM (both tibias and femurs), or spleen. For analysis of stem and progenitor cells, c-kit$^+$ cells were selected using anti-mouse CD117 microbeads or Lin$^-$ cells were selected using Lineage depletion microbeads (both from Miltenyi Biotec, San Diego, CA). The c-kit$^+$ or Lin$^-$ cells were stained with mouse antibodies (Supplementary Table 4) for further sorting or analysis. Stem and progenitor subpopulations were identified as LSK (Lin$^-$Sca-1$^{hi}$c-Kit$^{hi}$), multipotent progenitors (MPP) [i.e., LSK Flt3$^-$CD150$^-$CD48$^-$ (MPP1), LSK

Flt3$^-$CD150$^+$CD48$^+$ (MPP2), LSK Flt3$^-$CD150$^-$CD48$^+$ (MPP3) and LSK Flt3$^+$CD150$^-$ lymphoid-primed MPP (LMPP)] and long-term hematopoietic stem cells (LT-HSC; LSK Flt3$^-$CD150$^+$CD48$^-$)[27,29]. Myeloid progenitors were identified as Lin$^-$Sca-1$^-$c-Kit$^+$CD34$^+$FcγRII/III$^{lo}$ (CMP), Lin$^-$Sca-1$^-$c-Kit$^+$CD34$^+$FcγRII/III$^{hi}$ (GMP), or Lin$^-$Sca-1$^-$c-Kit$^+$CD34$^-$FcγRII/III$^{lo}$ (MEP)[70]. All analyses were performed on a Fortessa x20 flow cytometer (BD Biosciences) and sorting was performed on Aria Fusion instrument (BD Biosciences) and data were analyzed by BD FACSDiva or FlowJo software.

### Cell Culture
Human HPC (Lin$^-$CD34$^+$) and HSC (Lin$^-$CD34$^+$CD38$^-$) were cultured in Stemspan serum-free medium II (SFEM II, StemCell Technologies), supplemented with low concentrations of GFs similar to those present in long-term BM culture stroma-conditioned medium [granulocyte-macrophage colony-stimulating factor (GM-CSF) 200 pg/mL, leukemia inhibitory factor (LIF) 50 pg/mL, granulocyte colony-stimulating factor (G-CSF) 1 ng/mL, stem cell factor (SCF) 200 pg/mL, macrophage-inflammatory protein-1α (MIP-1α) 200 pg/mL, and interleukin-6 (IL-6) 1 ng/mL][71]. Murine BM LSKs were cultured in SFEM II supplemented with 10 ng/ml SCF and 10 ng/ml TPO. All the cells were cultured at 37 °C with 5% CO$_2$ and high humidity.

### Gene expression by Q-RT-PCR
To measure the miRNA and mRNA expression, total RNA was extracted using the miRNeasy Mini Kit (Qiagen, Valencia, CA). For miRNA expression, reverse transcription using MultiScribe™ Reverse Transcriptase and Q-PCR analysis using Taqman assays (Applied Biosystems) were performed according to the manufacturer's protocol. RNU44 and snoRNA234 was used as internal controls for human and mouse miRNA respectively. For mRNA expression, first-strand cDNA was synthesized using the SuperScript III First-Strand Kit and then Q-PCR was performed using TaqMan Gene Expression Assays (Applied Biosystems, Supplementary Table 4) or SYBR green PCR master mix and primers (ThermoFisher, Supplementary Table 4). BCR-ABL expression in human and mouse samples were measured with primer and probe sequences for BCR-ABL (B3A2 or B2A2), as previously described[72]. B2M and GAPDH were used as internal controls and the results are presented as log2-transformed ratio according to the 2$^{-\Delta Ct}$ method (ΔCt = Ct of target − Ct of reference).

### DNA sequencing
BM LSKs from miR-142$^{+/+}$BCR-ABL (CP CML), miR-142$^{-/-}$BCR-ABL (BC CML) (BCR-ABL was induced for three weeks by tet-off), miR-142$^{+/+}$ (WT) and miR-142$^{-/-}$ (KO) mice were sorted. Whole Genome sequencing libraries were prepared with Kapa HyperPrep kit (Kapa Biosystems) according to the manufacturer's protocol. The sequencing was

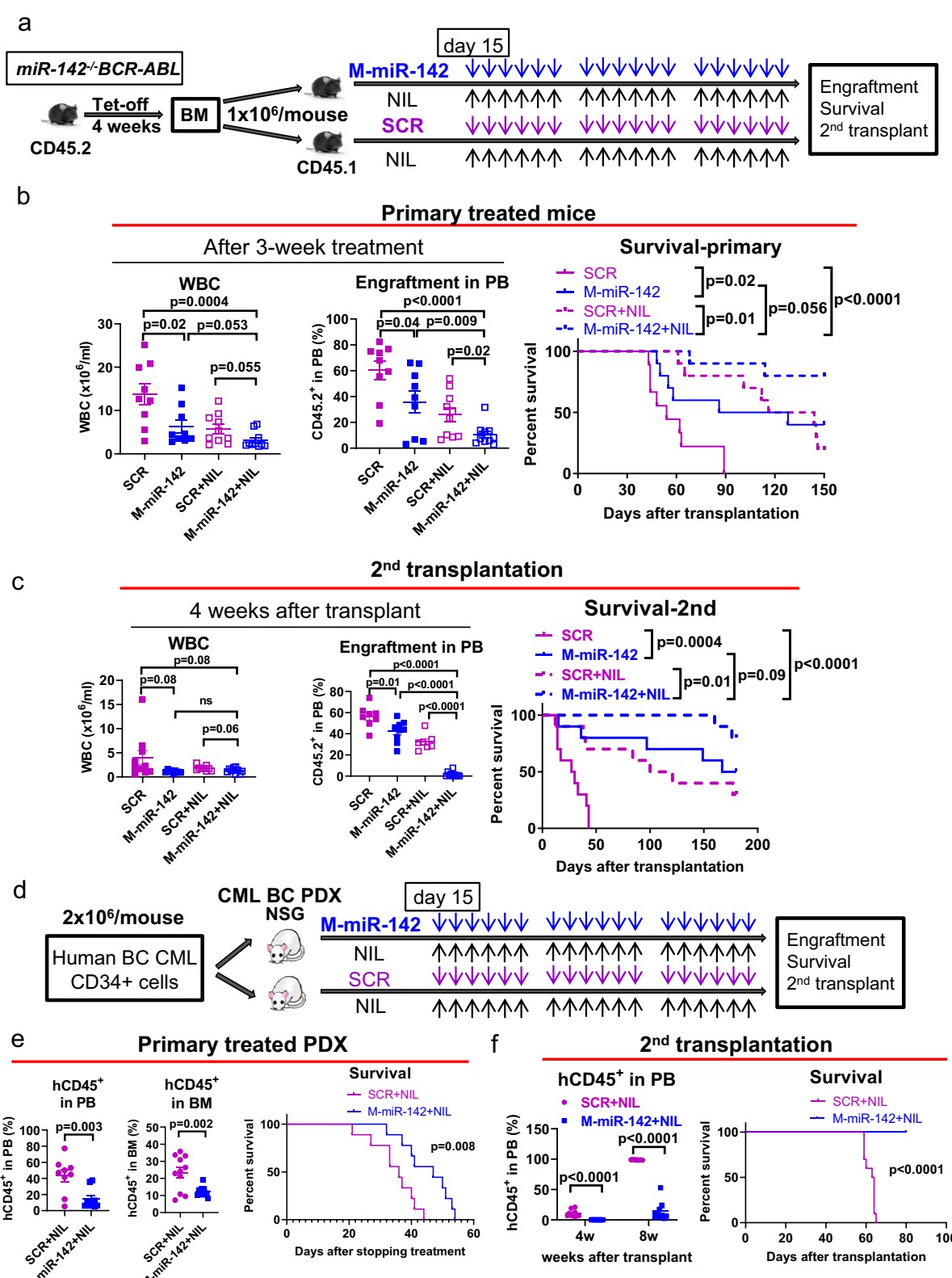

performed on Illumina NovaSeq6000 at Tgen with the sequencing length of 2x151. Filtered reads were aligned to the mouse reference genome (mm10) by Burrows-Wheeler Aligner v0.7.17 and PCR duplicates were removed by Picard v2.21.1. Local realignment around insertion/deletion (INDEL) sites and base quality recalibration (BQSR) were performed using GATK v4.1.7.0. To detect somatic copy number variation (CNV), we followed the best practices workflows of GATK somatic CNVs, using 1000 bp interval.

### Single cell RNA sequencing

BM LSKs from *miR-142^-/- BCR-ABL* (BC CML) and *miR-142^+/+ BCR-ABL* (CP CML) mice (BCR-ABL was induced for two week by tet-off) were sorted and ~8000 cells per sample were captured on a 10x Genomics Chromium using a Next GEM Single Cell 3' GEM kit V3.1 (10xGenomics). All protocols were performed following the manufacture's instruction. All samples have a viability of cells > 80%. Sequencing was performed on Illumina NovaSeq6000 at TGen with the sequencing length of

**Fig. 7 | CpG-M-miR-142 sensitizes BC CML to TKIs. a–c** Experimental design and results. **a** A cohort of BC CML mice were generated by transplanting $10^6$ BM MNCs from diseased *miR-142⁻/⁻ BCR-ABL* mice (CD45.2) into congenic wt recipient mice (CD45.1). At 2 weeks post transplantation, these mice were treated with CpG-M-miR-142 (20 mg/kg/day, IV), NIL (50 mg/kg/day, gavage), CpG-M-miR-142 + NIL, or SCR + NIL for 3 weeks. **b** WBC counts, PB engraftment rates, and survival (n = 10 mice per group). **c** WBC counts, PB engraftment rates, and survival of the 2nd recipient mice transplanted with BM MNCs from the treated donors (n = 10 mice per group). **d–f** Experimental design and results. **d** BC CML PDX mice were generated by transplanting $2 \times 10^6$ CD34⁺ cells from BC CML patients into NSG mice.

Upon detecting >5% human CD45⁺ cell engraftment in PB, these PDX mice were treated with CpG-M-miR-142 (20 mg/kg/day, iv) + NIL (50 mg/kg/day, oral garage) or SCR + NIL for 3 weeks. **e** PB and BM **e**ngraftment rates and survival (n = 10 mice per group). **f** PB engraftment rates and survival of the 2nd recipients transplanted with BM MNCs from the treated donors (n = 10 mice per group). For **b**, **c** and **e**, **f**, source data are provided as a Source Data file. M-miR-142 CpG-M-miR-142, NIL Nilotinib, SCR CpG-scramble miRNA control, WBC white blood cell, PB peripheral blood. Comparison between groups was performed by two-tailed, unpaired t-test. Survival curves were compared by Log-rank test. Results shown represent mean ± SEM.

28 + 10 + 10 + 10]. FASTQs were aligned using Cell Ranger v6.1.2, and count matrices were input to Seurat (v4.1.0) for subsequent analysis. Cells with low UMI counts (≤300) and high mitochondrial RNA content (≥10%) in each sample were discarded. Data normalization, integration, dimension reduction and cell clustering were conducted using "Seurat". The cell clusters were annotated based on expression patterns of known markers shown in Supplementary Fig. 3b, c. Differential gene expression was performed in each cluster of *miR-142⁻/⁻ BCR-ABL* LSKs vs *miR-142⁺/⁺ BCR-ABL* LSKs using "Seurat", and genes were ranked according to log2FC value. GSEA pathway analysis was then performed on the pre-ranked gene list by clusterProfiler (v4.2.2) on hallmark pathways.

## Bulk RNA sequencing

BM LSK cells from *miR-142⁺/⁺ BCR-ABL* (CP CML), *miR-142⁻/⁻ BCR-ABL* (BC CML) (BCR-ABL were induced for two weeks by tet-off), *miR-142⁺/⁺* (WT) and *miR-142⁻/⁻* (KO) mice (n = 5 per group) were sorted, and total RNA was extracted using the miRNeasy Micro Kit (Qiagen, Valencia, CA). RNA sequencing libraries were prepared with Kapa mRNA HyperPrep kit (Kapa Biosystems) according to the manufacturer's protocol. RNA-seq libraries were sequencing on Illumina NovaSeq6000 with the sequencing length of 2x101bp. RNA-Seq reads were trimmed to remove sequencing adapters using Trimmomatic and polyA tails using FASTP. The processed reads were mapped to the mouse genome mm10 using STAR (v. 2.6.0.a), and gene expression levels were summarized by HTSeq-count v.0.11.1. Gene expression raw counts were normalized using TMM normalization method and differential expression analysis was conducted using log likelihood ratio test implemented in "edgeR". DEGs between BC vs CP and between miR-142 KO vs WT were considered significant with an FDR-adjusted p-value less than 0.05. Genes ranked according to their log2 fold change and p-values were then subjected to pre-ranked GSEA analysis using hallmark pathways. Heatmaps were generated using "pheatmap" package.

## State-space construction using SVD

Singular value decomposition (SVD) was used on the mean-centered log-normalized counts to construct the state-space. The singular values produced were used to construct a screeplot, and the "elbow" of the screeplot was identified to be the fourth singular value. The principal components (PCs), or singular vectors produced by the SVD, were used as lower dimensional representations of the data. PC1 and PC2 were identified as the components that best separated the four experimental conditions. The space was rotated (-34.4 degrees for the full transcriptome state-space; 25.6 degrees for the metabolism state-space) which maximized the separation between the CML and non-CML samples and alignment with the x-axis to create an interpretable state-space. The rotated state-space was used for all analyses. When constructing the metabolism state-space, SVD was performed on the mean-centered, log-normalized expression of the 655 EMHGS genes.

## State-space comparison using mutual information (MI)

For each gene, MI was computed with respect to the gene expression and the experimental conditions as follows: $I(X : Y) = H(X) + H(Y) - H(X,Y)$ where $H(X)$ is the marginal entropy and $H(X,Y)$ is the joint entropy. Gene expression values were discretized into three bins; the boundaries of the three bins were determined using max entropy binning of the mean-centered log-normalized gene expression of all genes. The MI density was determined by comparing the distribution of MI between the genes used to construct the EMHGS state-space and the whole transcriptome state-space with the EMHGS genes omitted. To determine whether there was a difference in the information density, the Wilcoxon rank sum test was used to compare the distributions. The information density was compared for three experimental conditions: all four conditions (wt, miR-142 KO, CP, and BC); CP vs BC; or wt vs miR-142 KO.

## Metabolomic study

BM Lin⁻c-Kit⁺ cells, a fraction where LSKs reside, were selected from *miR-142⁻/⁻ BCR-ABL* and *miR-142⁺/⁺ BCR-ABL* mice (BCR-ABL was induced for 4 weeks by tet-off). Briefly, Lin⁻ cells were selected using Lineage depletion microbeads and c-kit⁺ cells were then selected using anti-mouse CD117 microbeads (both from Miltenyi Biotec, San Diego, CA). Since at least $3 \times 10^6$ cells per sample were necessary for metabolomic studies and we could not obtain a sufficient number of BM Lin-c-Kit+ cells from individual miR-142⁻/⁻ BCR-ABL or miR-142⁺/⁺ BCR-ABL mice, we randomly combined BM Lin-c-Kit+ cells from 2-3 mice to reach the number of $3 \times 10^6$ cells per sample. So, we obtained a total of 9 samples from 24 miR-142⁻/⁻ BCR-ABL mice and 7 samples from 18 miR-142⁺/⁺ BCR-ABL mice, which were then analyzed for metabolomics. For metabolite extraction, $3 \times 10^6$ pooled Lin-c-Kit+ cells were re-suspended into methanol: acetonitrile: water (2:1:1, v/v/v) containing four internal standards ($d_8$-Valine, $^{13}C_6$-Phenyl alanine, $^{13}C_6$ -adipic acid, $d_4$-succinic acid). The samples were vortexed for 30 s and subjected to three 30 sec freeze-thaw cycles for cell lysis. Cell lysates were then centrifuged at 15,000 rpm for 10 min at 4 °C and supernatants were split into two aliquots. One aliquot was directly used for HILIC LC-MS (hydrophilic interaction liquid chromatography–mass spectrometry) analysis; the second aliquot was evaporated to vacuum-dried at room temperature. Dried metabolite extracts were resuspended into 80% Water, 20% acetonitrile (*v/v*) for reversed phase (RP) LC−MS analysis. System suitability was determined using a plasma metabolite extract. A pool of samples (pooled QC) was generated by pooling 10 μL from each sample for RP and HILIC LC-MS. These pooled QC samples were injected before each batch for system conditioning, every five samples to determine instrument variability and to perform batch correction and normalization. Data acquisition was performed on an Ultimate 3000 RSLC with a HPG-3400RS binary pump coupled to Orbitrap Fusion Lumos Tribrid mass spectrometer (ThermoFisher, San Jose, CA) using HILIC LC-MS and RP LC−MS in both positive and negative ionization modes. The BEH amide (130 Å, 1.7 μm, 2.1 mm X 100 mm) and Hypersil GOLD C18 (175 Å, 1.9 μm, 2.1 × 150 mm) columns (Thermo-Fisher) were used for HILIC and RP chromatography, respectively. For HILIC chromatography, solvent A (95% water, 5% acetonitrile, 10 mM ammonium acetate, v/v/v) and solvent B (95% acetonitrile, 5% water, 10 mM ammonium acetate, v/v/v/) were used with a following gradient: 99% B for 1 min, 99–85% for 2 min, 85–75% B for 3 min, 75–30% B for 3 min, 30% B for 1 min. Finally, the column was re-equilibrated at initial conditions for 5 min at a flow rate of 0.4 mL/min and a column

temperature 45 °C. A 15 min RPLC chromatography was conducted using a 10 min linear gradient from 100% solvent A (water, 0.1% formic acid, v/v) and 0% solvent B (acetonitrile, 0.1% formic acid, v/v) to 2% solvent A and 98% solvent B, and 5 min of equilibration time at 0.35 mL/min flow rate were used for metabolite separation. MS1 data were acquired over a mass range of 70–1500 m/z in both positive and negative ionization modes in the orbitrap operated at a resolution of 60,000. MS/MS data were acquired in AcquireX mode using the iterative precursor exclusion workflow, with a stepped HCD collision energy at 20, 35 and 50, at a resolution of 30,000 in the orbitrap.

Data quality was determined by monitoring the performance of 20 amino acids in system suitability and pooled QC samples across four chromatographies. Areas of More than 75% of these analytes were within 25% RSD and the retention time deviation and mass error were within ±2 min and ±5ppm, respectively[46,73]. The area of at least one internal standard across all samples was within 10% RSD across all chromatographies, indicating that sample processing variability was within acceptable limits[46,73]. Raw data were subjected to Compound Discoverer 3.2.0.421 (ThermoFisher) for metabolite identification using HMDB, KEGG, LipidMaps and mzCloud databases, and relative quantitation. Any metabolites with an area under 25% RSD, a retention time deviation within ±2 min and a mass error within ± 5ppm were selected for downstream data analysis. Each metabolite was given a unique identifier. All four chromatographies were then combined, and normalized metabolite areas were used for statistical analysis. For the WT vs KO comparison, a two tailed t-test was performed, followed by Benjamini-Hochberg correction for multiple testing. An ANOVA was performed to determine differentially abundant metabolites across KO versus WT, and males and females, with a Tukey post-hoc test adjusted for multiple testing (Benjamini-Hochberg). Only annotated non-redundant endogenous metabolites were used for data representation and pathway analysis.

## ChIP assay
Chromatin immunoprecipitation (ChIP) was performed as described by the manufacturer (Pierce). Precleared chromatin was incubated overnight by rotation with 4 μg of NRF2 antibody or IgG antibody as a negative control. Immunoprecipitates were resuspended in 50 μL TE buffer. Inputs and immunoprecipitated DNA samples were quantified by q-PCR on a 7900T Fast real-time PCR system (Applied Biosystems). Primers are listed in Supplementary Table 4.

## FAO assay
Cells were first rinsed with HBBS before being treated with a 200 μL mixture of [3H]-palmitic acid (1 mCi/mL, Perkin Elmer) linked to fatty-acid free albumin (100 μM; maintaining a palmitate to albumin proportion of 2:1) and 1 mM l-carnitine. This complex was then left to incubate for a duration of 2 h at 37 °C. Subsequently, the supernatant was retrieved and added to a tube filled with 200 μL of chilled 10% trichloroacetic acid. The tubes then underwent centrifugation for 10 min at 3000 × g at 4 °C, after which aliquots of the supernatants (350 μL each) were collected and neutralized by adding 55 μL of 6 N NaOH. The solution was then transferred onto an ion exchange column filled with Dowex 1X2 chloride form resin (Sigma Aldrich, St. Louis, MO). The radioactive constituent was then extracted with water. Finally, the flow-through was collected and the radiation level was determined using liquid scintillation counting.

## Seahorse assay
Each well of a Seahorse XF-96-well cell culture microplate was planted with 100,000 cells in 180 μl of cell culture medium and incubated overnight at a temperature of 37 °C in a 5% $CO_2$ environment. For control purposes, three wells were left empty of cells and were supplemented solely with Seahorse media, consisting of basal XF media, glucose at 5.5 mM, sodium pyruvate at 1 mM, and glutamine at 4 mM

(furthermore, the pH was adjusted to 7.4). A Seahorse sensor cartridge was conditioned twelve hours before using the plate by placing it in a Seahorse Calibrant solution as per the guidelines given by the manufacturer, inside a $CO_2$-free incubator at 37 °C. On the day the assay was to be conducted, the cells were washed and treated with Seahorse media. The sensor cartridge was then mounted onto the cell culture plate and subsequently stored in a CO2-free incubator at 37 °C for sixty minutes. The assay was performed on the Seahorse XF96 Analyzer (Agilent, Santa Clara, CA) and involved the sequential injection of the following inhibitors as per the standard Cell Mito Stress Test: oligomycin (1.5 μM), FCCP (1 μM), and a combination of Rotenone/Antimycin A (0.5 μM).

## ATP assay
Intracellular ATP concentrations were measured using a colorimetric ATP assay kit (Abcam, #ab83355). Mouse LSK cells from *miR-142−/− BCR-ABL* and *miR-142+/+ BCR-ABL* mice, and CD34+CD38− cells from CP CML and BC CML patients, with or without CpG-M-miR-142 (2 μM) treatment for 24 h, were harvested, and ATP assay was performed according to the manufacturer's instructions.

## Immunoprecipitation and immunoblotting analysis
Mouse Lin−c-Kit+ or LSK cells from *miR-142−/− BCR-ABL* mice and *miR-142+/+ BCR-ABL* mice, and CD34+ or CD34+CD38− cells from CP CML and BC CML patients, with or without CpG-M-miR-142 (2 μM) treatment, were washed in ice-cold PBS and subsequently lysed in buffer containing 1 mM phenylmethanesulfonylfluoride and 10 mM protease inhibitor cocktail. For immunoprecipitation, 500 μg of cell lysate was incubated with indicated antibodies overnight at 4 °C. 30 μl of protein A/G agarose beads (Calbiochem) were added, and the mixture was inverted for 2 h at 4 °C. For immunoblotting, the immunoprecipitated complex or 30 μg of each cell lysate was separated on NuPAGE 4–12% gradient gels (Invitrogen) and immunocomplexes were visualized with enhanced chemiluminescence reagent (Thermo Scientific, Lafayette, CO). List of antibodies used for IP and IB analysis are shown in Supplementary Table 4.

## Cellular fractionation
The cells were washed with PBS, then divided into nuclear and cytoplasmic fractions utilizing a subcellular fractionation kit (Thermo Scientific, Lafayette, CO). The cells were vortexed at maximum speed in cytoplasmic extraction solution and then centrifuged to segregate the soluble cytoplasmic fraction. Nuclear extraction solution was added to the insoluble fraction (which contains the nuclei) and centrifuged to retrieve the nuclear fraction. Each step was carried out at 4 °C.

## Measurement of ROS
For determination of the level of ROS, the cells were incubated with 3 μM of MitoSOX Red (Life Technologies - Molecular Probes) for mitochondrial ROS analysis in culture media for 30 min at 37 °C. Then, cells were washed with PBS, and stained for Annexin V/DAPI analysis by flow cytometry. Only live cells population (Annexin-V-/DAPI-) was analyzed for ROS production.

## Transmission electron microscopy
Cultured cells were fixed with 2.5% glutaraldehyde, 0.1 M cacodylate buffer (Na(CH₃)₂AsO₂ ·3H₂O), pH7.2, at 4 °C. Standard sample preparation for TEM was followed including post-fixation with osmium tetroxide, serial dehydration with ethanol, and embedment in Eponate. Ultra-thin sections (70 nm thick) were acquired by ultramicrotomy, post-stained, and examined on an FEI Tecnai 12 transmission electron microscope equipped with a Gatan OneView CMOS camera. TEM images were taken at nominal ×11,000 magnification.

## Measurement of mitochondrial membrane potential

Mitochondrial membrane potential was visualized in treated cell stained with JC-1 (Cat# T3168, ThermoFisher) using a confocal microscope (LSM880, Zeiss). Cells were collected, washed in ice-cold PBS and mounted on glass slides using a Cytocentrifuge (CytoSpin4, 600 rpm, 10 min). Cells were then washed with PBS, fixed in 4% paraformaldehyde for 15 min and permeabilized in 0.5% Triton X-100 for 15 min. Then the cells were stained with JC-1 dye for 1 h at 37 °C. According to the manufacture, JC-1 dye exhibits potential-dependent accumulation in mitochondria, indicated by a green fluorescence emission at (~529 nm) for the monomeric form of the probe, which shifts to red (~590 nm) with a concentration-dependent formation of red fluorescent J-aggregates. Consequently, mitochondrial depolarization is indicated by a decrease in the red/green fluorescence intensity ratio (ThermoFisher). Quantification of fluorescence intensity was done using Zen Blue program (Zeiss).

## Lentiviral transduction of human CD34+ cells

GFP-expressing miRZip anti-miR-142-3p (CS940MZ-1, a custom order from System Biosciences, with H1 promoter for anti-miR-142-3p and EF1a promoter for GFP-T2A-Puro expression) and nontarget control (cat: SI506A-1, from System Biosciences) lentiviruses were produced and used for transduction of human HSCs. Briefly, human CD34+ cells were cultured overnight in SFEM II supplemented with IL-3 (25 ng/ml), IL-6 (10 ng/ml), SCF (50 ng/ml), TPO (100 ng/ml) and Flt-3 ligand (100 ng/ml). The next day, cells were resuspended in SFEM II and lentiviral supernatant [multiplicity of infection (MOI) = 30], supplemented with the above growth factors and 1xTransDux virus transduction reagent (System Biosciences), and centrifuged at 1500 × g for 90 minutes for transduction by spinoculation. We observed 50–80% of GFP+ cells in human CD34+ cells transduced with miR-142 KD lentivirus (MOI = 30) at 48 h.

## Oligodeoxynucleotide design and synthesis

To synthesize CpG- M-miR-142, we modified our scavenger receptor (SR)/Toll-like receptor 9 (TLR9)–targeting platform for the transfer of 25/27-mer dicer-substrate small interfering RNA (siRNA) and miRNA inhibitors[28,69]. The double-stranded sequence of miR-142 was conjugated through the 5′end of the passenger strand to the 3′end of a single-stranded, partly phosphorothioated oligodeoxynucleotide (CpG ODN) using a synthetic carbon linked to obtain CpG-M-miR-142. To ensure the maximum activity and target specificity and to confer nuclease resistance to CpG-M-miR-142, the miR-142 moiety was also 2′-O-methyl modified at the 3′end of the passenger strand[68]. The partially phosphothioated ODN and miR-142 passenger or scrRNA was linked using 5 units of C3 carbon chain linker, (CH2)3 (indicated by x in the sequences below). The constructs were also conjugated with Cy3 to track the internalization in the cells by flow cytometry. The sequences were as follows:

**miR-142-3p mimic (guide):** 5′- rUrGrU rArGrU rGrUrU rUrCrC rUrArC rUrUrU rArArG rGrA -3′; **CpG-miR-142-3p mimic (passenger):** 5′- G*G*T GCA TCG ATG CAG G*G*G* G*G xxxxx rC rArUrA rArArG rUrUrG rGrArA rArCrA rCrUrA rCrAmA rA-3′; **Scramble RNA (scrRNA, guide):** 5′-rGrGrCrGrUrGrUrArUrUrArArGrGrCrUrArArArUrCrU-3′; **CpG-scrRNA (passenger):** 5′-G*G*T GCATCGATGCAGG*G*G*G*G xxxxx rArUrUrUrArGrCrCrUrUrArArUrArCrArCrGrCrCmArA-3′, where 'r' indicates ribo, '*' indicates phosphorothioation, one nonbridging atom of oxygen on phosphate was replaced with sulfur, 'x' indicates a C3 Spacer, and 'm' indicates the 2′-O-methyl analog of the nucleotide. The annealed miR-142-3p is as follows:

```
        CpG D19 ODN              linker      Passenger strand miR-142a-3p
5′ GGTGCATCGATGCAGGGGGG-o-o-o-o-o-CAUAAAGUAGGAAACACUACAAA 3′
                     3′ AGGUAUUUCAUCCUUUGUGAUGU 5′
                        Guide strand miR-142a-3p
```

## Apoptosis, cell growth and colony-forming cell assays

After transduction performed as above, GFP+ cells selected at 48 h were analyzed for apoptosis, cell growth, and colony forming cell (CFC) assay. An aliquot of GFP+ cells were exposed to nilotinib (NIL, Novartis) for 72 h and analyzed for apoptosis and cell growth. Human HSCs (i.e., CD34+CD38−) from CP CML or BC CML patients and murine LSCs (i.e., LSKs) from *miR-142+/+BCR-ABL* (CP CML) or *miR-142−/−BCR-ABL* (BC CML) mice were also treated with CpG-M-miR-142 or CpG-scrRNA (2 μM), with or without NIL (2 μM), for 72 h and analyzed for apoptosis and cell growth. Apoptosis was measured by labeling cells with Annexin V-PE or FITC or APC and 4, 6-diamidino-2-phenylindole (DAPI, BD-PharMingen, San Diego, CA) and analyzed by flow cytometry. Cell growth was measured by Lumino Glo (Promega). For CFC, miR-142 KD CD34+ cells or control cells (transduced with nontarget lentivirus) were plated in methylcellulose progenitor culture and burst-forming unit-erythroid and colony-forming unit-granulocyte and macrophage cells were counted after 14 days.

## In vivo treatment of mice

To determine the impact of in vivo rescue of miR-142 deficit by CpG-M-miR-142 on the BC transformation rate, a cohort of *miR-142−/−BCR-ABL* mice were treated with CpG-M-miR-142 (20 mg/kg/day, iv) or SCR for 4 weeks, starting on the day after tet-off BCR-ABL induction. At the completion of treatment, circulating blasts and LSKs and survival were compared between these two groups. To assess the impact of CpG-M-miR-142 treatment on the LSC burden, we then transplanted 10⁶ BM cells from CpG-M-miR-142- or SCR-treated mice (CD45.2) into secondary (2nd) recipient mice (CD45.1). Engraftment rates and survival were monitored in the 2nd recipients.

To determine the in vivo antileukemic efficacy of CpG-M-miR-142 on a model where BC transformation had already occurred, a cohort of *miR-142−/−BCR-ABL* mice were treated with CpG-M-miR-142 (20 mg/kg/day, iv) or SCR, starting on day 15 after tet-off BCR-ABL induction, for 3 weeks. Leukemic blasts and survival were monitored. Secondary transplant experiment was also performed to evaluate post-treatment LSC burden. To validate results in the human disease, we generated a patient-derived xenograft (PDX) by transplanting 10⁶ CD34+ cells from BC CML patients into NSG mice. Upon detecting engraftment (>5% blood human CD45+ cells), we treated the animals with CpG-M-miR-142 (20 mg/kg/day, iv) or SCR for 3 weeks. A cohort of mice were monitored for survival and another cohort were used for post-treatment evaluation of residual LSC burden by transplanting BM cells from these treated mice into 2nd NSG recipient mice. Donor engraftment rates and survival were monitored.

To test in vivo whether rescuing miR-142 deficit increases TKI sensitivity of BC cells, a cohort of BC CML mice were generated by transplanting 10⁶ BM MNCs from diseased *miR-142−/−BCR-ABL* mice (CD45.2) into congenic wt recipient mice (CD45.1). At 2 weeks post transplantation, these mice were treated with CpG-M-miR-142 (20 mg/kg/day, IV), NIL (50 mg/kg/day, gavage), CpG-M-miR-142 + NIL, or SCR + NIL for 3 weeks. Leukemic burden and survival were monitored. Secondary transplant experiment was also performed to evaluate post-treatment LSC burden. A cohort of BC CML PDX mice were also generated by transplanting 10⁶ CD34+ cells from BC CML patients into NSG mice. Upon detecting >5% human CD45+ cell engraftment in PB, we treated the PDX mice with CpG-M-miR-142 (20 mg/kg/day, iv) + NIL (50 mg/kg/day, oral gavage) or SCR + NIL for 3 weeks and monitored survival of these mice. Secondary transplant experiment was also performed to evaluate post-treatment LSC burden. Human CD45+ cell engraftment rates and survival were monitored in the 2nd NSG recipient mice.

## Statistics & reproducibility

All statistical analyses were performed using Prism version 8.0 software (GraphPad Software). For all animal experiments, numbers of

biologically independent mice were provided. For immunoblotting experiments, results from one of three independent experiments are shown. For all in vitro experiments, numbers of biologically independent samples used in each experiment are provided. Comparison between two groups was examined by one-tailed or two-tailed, unpaired t-test. The log-rank test was used to assess significant differences between survival curves. No statistical method was used to predetermine sample size. No data were excluded from the analyses. Randomization was used for all animal experiments. For some experiments, investigators were blinded to mouse genotype while performing treatment or monitoring for engraftment or survival. For the remaining experiments, the investigators were not blinded to allocation during experiments and outcome assessment. For all cases, statistical significance was set as $p < 0.05$. Results shown represent mean ± standard error of the mean (SEM) or standard deviation (SD), as indicated. *$p < 0.05$; **$p < 0.01$; ***$p < 0.001$; ****$p < 0.0001$; ns not significant.

### Reporting summary

Further information on research design is available in the Nature Portfolio Reporting Summary linked to this article.

## Data availability

The DNA, RNA and scRNA sequencing data generated in this study are available at the Sequence Read Archive (SRA) or the Gene Expression Omnibus (GEO) repository of the National Center for Biotechnology Information under accession code PRJNA895411, GSE216794 and GSE217076. Metabolomic profiles are available at the NIH Common Fund's National Metabolomics Data Repository (NMDR) website, the Metabolomics Workbench[74] (https://www.metabolomicsworkbench. org, Study ID ST002446, Project https://doi.org/10.21228/M8VD8T). Supplementary information, including Supplementary Figs. 1–10 and Supplementary Tables 1–4, are provided with the online version of this paper. Additional Supplementary files, including Supplementary Data 1 and 2 are provided with the online version of this paper. Source data are provided with this paper.

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

## Acknowledgements

This work was supported in part by the Robert & Lynda Altman Family Foundation Research Fund, Borstein Foundation, and National Cancer Institute grants: CA258981(GM/BZ), CA248475 (GM/BZ), CA205247

(YHK/GM), CA25004467 (RCR/YHK/GM). We acknowledge the support of the Animal Resources Center, Analytical Cytometry, Pathology (Liquid Tumor), Light Microscopy, Electron microscopy, Integrated Mass Spectrometry, Integrative Genomics, and DNA/RNA Shared Resources at City of Hope Comprehensive Cancer Center supported by the National Cancer Institute of the National Institutes of Health under award number P30CA33572. We are grateful to the City of Hope Comprehensive Cancer Center, the patients, and their physicians for providing primary patient material for this study.

## Author contributions

B.Z. designed and conducted experiments, analyzed data, wrote manuscript, and provided administrative support; D.Z., F.C., H.W., K.P., A.T., K.G.M., Y.Z., and D.H.H. conducted experiments and analyzed data; D.F., L.D., M.H.C., S.T., H.C., S.B., R.R., X.W., and R.S. analyzed DNA-seq, RNA-seq and scRNA-seq; L.G. provided patient samples; M.B. provided miR-142 KO mice and reviewed the manuscript; Y.L., P.S. and M.K. designed and manufactured CpG-M-miR-142; D.P., L.L., J.J., J.C., J.Y., M.C., Y.H.K., and P.P. reviewed data and the manuscript; L.X.T.N. designed experiments, analyzed data, and wrote manuscript; G.M. designed experiments, analyzed data, wrote manuscript, and provided administrative support.

## Competing interests

The authors declare no competing interests.

## Additional information

[1]Department of Hematological Malignancies Translational Science, Gehr Family Center for Leukemia Research, City of Hope Medical Center and Beckman Research Institute, Duarte, CA, USA. [2]Department of Computational and Quantitative Medicine, City of Hope Medical Center and Beckman Research Institute, Duarte, CA, USA. [3]Department of Hematology, the First Affiliated Hospital, College of Medicine, Zhejiang University, Hangzhou, Zhejiang, PR China. [4]Cancer & Cell Biology Division, Translational Genomics Research Institute, Phoenix, AZ, USA. [5]Integrated Mass Spectrometry Shared Resource, City of Hope Comprehensive Cancer Center, Duarte, CA, USA. [6]Department of Systems Biology, Beckman Research Institute of City of Hope, Monrovia, CA, USA. [7]City of Hope National Medical Center, Integrative Genomics Core, Department of Computational and Quantitative Medicine, Beckman Research Institute, Duarte, CA, USA. [8]DNA/RNA Peptide Shared Resources, Beckman Research Institute, Duarte, CA, USA. [9]Department of Medicine and Greenebaum Comprehensive Cancer Center, University of Maryland School of Medicine Baltimore, Baltimore, MD, USA. [10]Department of Immunology and Inflammation, Centre of Hematology, Imperial College of London, London, UK. [11]Department of Hematology & Hematopoietic Cell Transplantation, City of Hope National Medical Center, Duarte, CA, USA. [12]Department of Immuno-Oncology, Beckman Research Institute, Duarte, CA, USA. [13]These authors contributed equally: Bin Zhang, Dandan Zhao, Fang Chen. [14]These authors jointly supervised this work: Bin Zhang, Le Xuan Truong Nguyen, Guido Marcucci. ✉e-mail: bzhang@coh.org; lenguyen@coh.org; gmarcucci@coh.org

