## [Peer Review File · Nature Communications]

Acquired miR-142 Deficit in Leukemic Stem Cells Suffices to Drive Chronic Myeloid Leukemia into Blast CrisisREVIEWER COMMENTS

Reviewer #1 (Remarks to the Author):

Zhang, Zhao, and Chen et al. examines the role of miR-142 in driving CML into blast phase. The authors demonstrate that decreased expression of miR-142, which they observe in blast phase CML patient specimens, results in a more aggressive leukemia in GEMMs. Mechanistically, the authors show that miR-142 loss increases OxPhos, fatty acid oxidation, and mitochondrial fusion while decreasing ROS levels. The use of both patient specimens and GEMM models is a real strength of this manuscript. Overall, this is a very intriguing manuscript with a lot of mechanistic data. Several things can be done to further improve on this manuscript, listed below.

1. Since loss of miR-142 results in numerous hematopoietic changes, it would be helpful to provide data related to the miR-142 +/- and -/- mice relative to the miR +/- B/A and -/- B/A models including spleen size, blood cell counts, and survival (currently summarized in a table format). Currently it is difficult to know if some of the phenotypes observed in the miR-142 -/- BA mice is combinatorial or driven by miR-142 loss.
2. Limiting dilution experiments to measure stem cell frequency require the examination of several cell doses from the same population. The description of Figure 2a and 2b should be reworded. This is experiment is very nice, but it is really identifying the population of leukemia initiating cells or leukemia stem cells which the authors show is in the LSK compartment.
3. Etomoxir has many off-target effects. To support this data, it would be helpful to use RNAi or genetic approaches to perturb CPT levels and measure the consequences on the leukemia using the same in vitro assays shown in Supplemental figure 4f.
4. In the Seahorse assays, the figure makes it appear that FCCP and Oligomycin were given at the same time but the methods state the drugs were given sequentially. Can the authors clarify this. If Oligomycin was given alone why wasn't a drop in OCR observed? If they were given at the same time, can the authors explain why this was done over the more standard mito-stress test.
5. The authors should quantify the mitochondrial phenotypes observed in Figure 5 to help support the representative images.
6. The authors show changes in LC3 levels. Do they observe changes in autophagy or mitophagy in the imaging studies? Are the changes in mitochondrial biology leading to changes in OxPhos and/or FAO?
7. In figure 7, additional statistics to measure single agent vs. combination therapy should be included (miR-142 vs. miR-142 + NIL).
8. Do the authors observe changes in ATP levels in the miR-142-/-, miR-142-/-BA, and miR+/+ BA cells or CpG-M-miR-142 treated cells?
9. Does blocking ROS induction by adding antioxidants to the cells upon CpG-M-miR-142 treatment rescue cell death?
10. Does etomoxir treatment/CPT knockdown result in changes in FAO and OXPHOS in these models?

Minor Comments:

1. The logic behind some statement including "Thus, low ROS levels identify "fit" LSCs" are hard to follow. The reference the authors are citing shows that functional LSCs have low ROS levels. Neither current manuscript nor referenced paper speaks to fitness of LSCs. This reviewer suggests removing statements like this because they detract from the nice data presented.
2. Several times in the manuscript the authors use the word validate. In some cases, it seems a little misleading since the data presented is not using a different technique to access the same biology but examining a complementary piece of biology. For example, the changes in ROS levels

do not validate the metabolomics data but they certainly support the metabolomics data.
3. It would be helpful to graph figure 6i so that the reader can appreciate the engraftment differences at both 4 and 8 weeks. The current figure only shows 8 weeks clearly.

Reviewer #2 (Remarks to the Author):

Zhang et al. reports a role of miR-142 in the blast crisis (BC) transformation of chronic myeloid leukemia (CML). miR-142 is expressed at lower levels in BC patients versus chronic phase (CP) CML. Knocking out miR-142 leads to BC phenotypes in a GEMM model of CML. Treatment with miR-142 mimetics leads to improved survival of mouse and human CML in murine models, and enhances the effects of a BCR-ABL inhibitor. The authors further described many downstream metabolic changes in miR-142 KO and patient BC samples, although many of these changes are likely secondary.

Overall, this is an interesting manuscript, because the mechanisms of BC transformation of CML is not well understood. The presented data are overall strong and of good quality. I have a few suggestions to further enhance the manuscript.

1. The authors claimed that WT BCR-ABL mice do not develop BC whereas miR-142^{-/-} mice do, which is a key point of the manuscript.
 - a. Fig 1e,1f and Fig 2e examined at the 4-week time point. It is important to also show PB and BM cell morphology for moribund mice for the three genotypes, to eliminate the possibility that the morphological differences at 4 weeks is simply due to a faster disease progression in miR-142^{-/-} and +/- mice.
 - b. Similarly, on top of Fig 6e, it will be interesting to know whether the mice still die from BC CML after miR-142 mimic treatment by examining cell morphology in moribund mice.
2. It has been previously shown that Musashi 2 is upregulated in BC CML and drives BC transformation. It is surprising that the authors did not mention MSI2. What is the relationship between MSI2 and miR-142? Does MSI2 suppress miR-142? Does miR-142 suppress MSI2? Or they are unrelated? Also, there are multiple published MSI2 CLIP datasets in hematopoietic cells that can be analyzed to see if MSI2 binds to the miR-142 region.
3. A substantial part of the manuscript is describing the metabolic and mitochondrial changes in miR-142^{-/-} CML. I can see the attempt by the authors to address potential mechanisms without identifying functional miR-142 targets. I recognize that it is not always feasible to identify a single target of a miRNA that explains most of its phenotypes. However, this descriptive part is too long and to some degree dilutes the more interesting messages on the biology of miR-142. I suggest the authors shorten these sections.

Minor:

1. For the single cell RNAseq analysis, it seems that the authors do not have data from more than one sample for WT vs miR-142 KO. The population size changes (Fig 2g, 2h) should therefore be only suggestive. Additionally, I am concerned about calling these sub populations from LSKs as HSCs, because there is no evidence that they all have stem cell properties. Calling them progenitors will be more accurate.
2. Figures 3h and 3i are misleading. It is a given that when comparing differentially expressed pathways vs the rest of the transcriptome, the per-gene information content will be higher in the pathway. This is because the rest of the transcriptome contains many unrelated or housekeeping genes. These data do not support that EMHGS are necessarily more important than other genes.
3. Fig 4b: the annotation of significance (the *) is too small, and hard to see.
4. Fig 6b: Please describe what red and blue lines are. If red is scrambled mimic, why there is no uptake of the scrambled?

Reviewer #3 (Remarks to the Author):

Chronic myeloid leukemia (CML) is normally diagnosed in chronic phase (CP) but develops into lethal blast crisis (BC) if left untreated, which is associated with additional mutations. In this manuscript the authors investigate the role of miR-142 in CP CML transitioning to BC CML. Overall the authors apply sc-RNA-seq, various types of metabolomic readouts and electron microscopy on a new mice model of miR-142 deficient CML and patient derived samples. For the latter, quite a large amount of work has been done with rare human LSCs (CD34+CD38-), which is impressive. The paper is well written, and the topic is of interest to the reader of Nature Communications.

Initially, the authors apply the mouse model and present convincing results showing that absence of miR-142 causes a BC phenotype. The authors performed sc-RNA-seq on mouse LSK (stem and progenitor cells) which suggested a clear metabolic effect, with an increase in central carbon metabolism in miR-142 depleted mice. Similar effect was seen when with untargeted metabolomics, where an increase in fatty acid oxidation was particularly evident. This was also confirmed by labelled palmitic acid and Seahorse assays, although some clarifications and additional experiments are needed in this part.

The authors correlated the increase in metabolic activity following miR-142 depletion with an upregulation of Mitofusin-1 (MFN1), enhanced mitochondrial fusion, decreased autophagy flux (LC3 levels) and decreased ROS levels. This part is preliminary/underdeveloped and mainly speculative. Further experiments are therefore required, with increased n-numbers and statistical analysis, to strengthen this part (i.e., while the electron microscopy looks clear, quantification or multiple representative images would support this).

Finally, the authors showed and expressing miR-142 mimic in vivo enhanced survival of mice in the mouse model and a BC PDX model.

Specific comments:

1. Figure 1 and Supplementary related figures 1-2: What are the absolute counts here for all cell types in spleen and bone marrow, based on total number of cells (cell counts). Maybe LT-HSCs are getting exhausted or if number of LSKs is high enough, a reduced % LT-HSCs won't translate to reduced total number.
2. Metabolomic analysis on page 12: How is the replicates plotted obtained if samples are pooled? How were they pooled, was it 2 mice to one sample or were all mice pooled and technical replicates made?
3. What pathway analysis was used for untargeted metabolomics?
4. Figure 5b: Figure legend suggests n=3. Quantification with statistical analysis would be useful.
5. Figure 5d-e: The Seahorse experiments require further explanation in text/figure legends. Why have the authors used Oligomycin and FCCP simultaneously? Here it is important to include antimycin/rotenone to shut down OCR and ascertain the differences in basal and maximal OCR are not simply due to different cell numbers. Statistical analysis is also required when interpreting the data.
6. Figure 5f-k: Quantification and statistical analysis are required to interpretate these data.
7. Supplemental Figure 6: n=2 or not stated for western blots (this should be in legends for all figures in general). Quantification and statistical analysis are required for key experiments.
8. Supplementary Figure 7h-i, l: Data regarding autophagy are not convincing. Is this LC3-I or lipidated LC3-II? Here autophagy flux should be measured using appropriate inhibitors (i.e. Bafilomycin) in at least one of the model (mouse or human). The levels of LC3 by Western alone

cannot accurately predict autophagy flux (the authors may consider measuring LC3 puncta by IF).

9. Supplementary Figure 7j: JC-1 experiment is not described well in the text or figure legends. Also, quantification and statistical analysis is required.

10. Figure 7: Are same samples used in Figure 6 h-I and 7 e-f (i.e., what is the effect on Nilotinib alone in figure 7 e-f)?

References: There has been limited work done on miR-142 in CML which should be referenced (for example: <https://www.frontiersin.org/articles/10.3389/fonc.2021.718731/full>). The authors cite the following paper (<https://www.jci.org/articles/view/123839>) but don't discuss that in dendritic cells, where miR-142 also regulates FAO. It would be useful to mention this a potential effect on normal or abnormal immune function.

Graphical abstract is underdeveloped and should be improved.

RESPONSE TO REVIEWERS' COMMENTS

Reviewers' comments:

Reviewer # 1 (Remarks to the Author):

C: Overall, this is a very intriguing manuscript with a lot of mechanistic data.

R: We thank the Reviewer for the positive comment.

C.1: Since loss of miR-142 results in numerous hematopoietic changes, it would be helpful to provide data related to the miR-142 +/- and -/- mice relative to the miR +/- B/A and -/- B/A models including spleen size, blood cell counts, and survival (currently summarized in a table format). Currently it is difficult to know if some of the phenotypes observed in the miR-142 -/- BA mice is combinatorial or driven by miR-142 loss.

R: We now provide the requested data on page 6 of the main text and in Supplementary Figures. 1 and 2 of the revised manuscript.

C.2: Limiting dilution experiments to measure stem cell frequency require the examination of several cell doses from the same population. The description of Figure 2a and 2b should be reworded. This experiment is very nice, but it is really identifying the population of leukemia initiating cells or leukemia stem cells which the authors show is in the LSK compartment.

R: We agree with the Reviewer and have reworded the description of Figures 2a and 2b (see page 8) accordingly.

C.3: Etomoxir has many off-target effects. To support this data, it would be helpful to use RNAi or genetic approaches to perturb CPT levels and measure the consequences on the leukemia using the same in vitro assays shown in Supplemental figure 4f.

R: We appreciate that ETO has potential off-target effects. This compound has been considered as a standard reagent to study CPT1 activity and, herein, it was used accordingly. Nevertheless, the Reviewer has a very good point. Thus, as suggested, we knocked down CPT1 with siRNAs and obtained results (i.e., increased apoptosis, reduced cell growth, and reduced CFC; Supplementary Fig. 8c) similar to those obtained by treating the cells with ETO (Supplementary Fig. 8a).

C.4: In the Seahorse assays, the figure makes it appear that FCCP and Oligomycin were given at the same time but the methods state the drugs were given sequentially. Can the authors clarify this. If Oligomycin was given alone why wasn't a drop in OCR observed? If they were given at the same time, can the authors explain why this was done over the more standard mito-stress test.

R: We thank the Reviewer for pointing this out. We initially utilized the "Seahorse XF Cell Energy Phenotype Test kit-Agilent cat #103325-100" to measure levels of OCR and ECAR. The Agilent Seahorse Cell Energy Phenotype Test kit is described by the manufacturer as a qualitative assay for determining cellular energy phenotypes. It simultaneously measures mitochondrial respiration

and glycolysis potential in live cells. For this assay, FCCP and Oligomycin were given at the same time as per manufacturer's instruction.

To address the metabolism questions of Reviewers' 1 and 3, we repeated these experiments using the "Seahorse XF Cell Mito Stress Test kit-Agilent cat #103010-100" and measured levels of OCR and ECAR in 1) mouse miR-142+/+ B/A vs miR-142-/- B/A LSK and human CP CD34+38- vs BC CD34+38- cells; and 2) mouse miR-142-/- B/A LSK and human BC CD34+38- cells treated with SCR control or M-miR-142.

Consistent with the previous results, we observed the increase of OCR but not ECAR in mouse miR-142-/- B/A LSK and human BC CD34+38- cells compared with mouse miR-142+/+ B/A LSK and human CP CD34+38- cells, respectively. M-miR-142 treatment significantly inhibited levels of OCR but did not change ECAR in both mouse miR-142-/- B/A LSK and human BC CD34+38- cells (see Figure 5d-e and Supplementary Figure 8g-j).

C.5: The authors should quantify the mitochondrial phenotypes observed in Figure 5 to help support the representative images.

R: Done. Please see Supplementary Figures 9g, 9i and 9m.

C.6: The authors show changes in LC3 levels. Do they observe changes in autophagy or mitophagy in the imaging studies? Are the changes in mitochondrial biology leading to changes in OxPhos and/or FAO?

R: As also suggested by Reviewer #3, we performed imaging studies of mouse miR-142-/- B/A LSK and human BC CD34+38- cells treated with SCR or M-miR-142 using immuno-fluorescence staining with LC3 antibody. Altogether these studies showed that miR-142 deficit associated with decreased autophagy. Conversely, treatment with M-miR-142 but not with SCR increased miR-142 levels and stimulated LC3 puncta formation, suggesting an alteration of the autophagy flux (Fig. 1). In order to correctly interpret these changes, as suggested by Reviewer 3, we also treated the cells with bafilomycin A1, a lysosomal inhibitor (Klionsky et al., *Autophagy*, 2016; Gump et al., *Autophagy*, 2014). The results seemingly confirmed that autophagy was increased after M-miR-142 treatment. since LC3 puncta formation increased after treatment with M-miR-142 and remained relatively unchanged after exposure to Bafilomycin A1 (Fig. 1). Of note, given that **a.** these results are somewhat tangential to the main theme of the manuscript; **b.** additional experiments may be necessary (see also response to Reviewer 3 C.8); and **c.** we have already reached space limitations, we have now elected to eliminate this part. We will be very willing to reinstate it if the Reviewers and the Editors wish so and space limitations permit.

Regarding the causative association of mitochondrial dynamics and metabolism functions (if we interpreted correctly what the Reviewer means for “mitochondrial biology”), our results demonstrated that miR-142 depletion increased concurrently both mitochondrial fusion and levels of FAO/OXPHOS while treatment of M-miR-142 reversed both of those effects (Fig. 5 and Supplementary Fig. S7). These results suggest a functional association between mitochondrial dynamics and levels of FAO/OXPHOS. To this end, others have previously reported that mitochondrial fusion supports OXPHOS, thereby maximizing the oxidative capacity of mitochondria in response to toxic stress (Yao et al. eLife, 2019; Youle and van der Bliek, Science, 2012). In addition, mitochondria fuse when cells are forced to rely more on OXPHOS (Rossignol et al., Cancer Research, 2004).

C.7: In figure 7, additional statistics to measure single agent vs. combination therapy should be included (miR-142 vs. miR-142 + NIL).

R: Done

C.8: Do the authors observe changes in ATP levels in the miR-142^{-/-}, miR-142^{-/-}BA, and miR^{+/+} BA cells or CpG-M-miR-142 treated cells?

R: Using an ATP assay kit (#Ab83355), we observed a significant increase in ATP levels in mouse miR-142^{-/-} B/A LSK and human BC CD34⁺CD38⁻ cells compared with mouse miR-142^{+/+} B/A LSK and human CP CD34⁺CD38⁻ cells, respectively. Conversely CpG-M-miR-142 treatment decreased ATP levels in mouse miR-142^{-/-}BA LSK and human BC CD34⁺CD38⁻ cells (see Supplementary Figure 9a-b).

C.9: Does blocking ROS induction by adding antioxidants to the cells upon CpG-M-miR-142 treatment rescue cell death?

R: To block ROS induction, we treated mouse miR-142^{-/-} B/A LSK and human BC CD34⁺38⁻ cells with CpG-M-miR-142 and/or *N*-acetyl-L-cysteine (NAC), a ROS scavenger. Flow cytometry assays demonstrated that NAC reversed the effects of CpG-M-miR-142 on ROS production and apoptosis (see Supplementary Figure 9f).

C.10: Does etomoxir treatment/CPT knockdown result in changes in FAO and OXPHOS in these models?

R: After treating mouse miR-142^{-/-} B/A LSK and human BC CD34⁺38⁻ cells with etomoxir (ETO, 5 μ M) or transfecting them with CPT1A/CPT1B siRNAs (20nM), we measured levels of FAO/OXPHOS. Both ETO and CPT1A/B siRNAs significantly inhibited FAO/OXPHOS compared with DMSO or SCR treatment, respectively (see Supplementary Figure 7m-o).

Minor Comments:

1. The logic behind some statement including “Thus, low ROS levels identify “fit” LSCs6” are hard to follow. The reference the authors are citing shows that functional LSCs have low ROS levels. Neither current manuscript nor referenced paper speaks to fitness of LSCs. This reviewer suggests removing statements like this because they detract from the nice data presented.

R: Done

2. Several times in the manuscript the authors use the word validate. In some cases, it seems a little misleading since the data presented is not using a different technique to access the same biology but examining a complementary piece of biology. For example, the changes in ROS levels do not validate the metabolomics data but they certainly support the metabolomics data.

R: Done

3. It would be helpful to graph figure 6i so that the reader can appreciate the engraftment differences at both 4 and 8 weeks. The current figure only shows 8 weeks clearly.

R: We re-graphed Fig. 6i as recommended by the Reviewer.

Reviewer #2 (Remarks to the Author):

Overall, this is an interesting manuscript, because the mechanisms of BC transformation of CML is not well understood. The presented data are overall strong and of good quality.

R: We thank the Reviewer for the positive comments.

Major comments:

C.1. The authors claimed that WT BCR-ABL mice do not develop BC whereas miR-142^{-/-} mice do, which is a key point of the manuscript.

a. Fig 1e,1f and Fig 2e examined at the 4-week time point. It is important to also show PB and BM cell morphology for moribund mice for the three genotypes, to eliminate the

possibility that the morphological differences at 4 weeks is simply due to a faster disease progression in miR-142^{-/-} and +/- mice.

R: PB and BM cell morphology of the moribund miR-142 KO BCR-ABL and miR-142 WT BCR-ABL mice are provided in **Supplementary Fig. 2c** of the revised manuscript. Since WT and miR-142 KO mice did not develop leukemia and had a long lifespan (> 600 days, see Supplementary Fig. 2a-b), we elected to use 40-week-old mice of these two strains for the requested comparison of PB and BM morphologies (see **Supplementary Fig. 2c** of the revised manuscript).

b. Similarly, on top of Fig 6e, it will be interesting to know whether the mice still die from BC CML after miR-142 mimic treatment by examining cell morphology in moribund mice.

R: Four-week treatment with miR-142 mimic rescued the BC phenotype, but once the treatment was discontinued, the treated mice eventually relapsed with BC. In secondary transplant experiments, we observed decrease of LSC burden as recipients of BM from miR-142 mimic-treated donors lived longer than recipients of BM from SCR-treated donors. We cannot exclude that a treatment of miR-142 mimic alone or in combination with TKI longer than 4 weeks could have led to even better results than those provided in the manuscript and perhaps eradicated LSCs. However, since miR-142 was given IV (either by retroorbital or tail vein injections), technical and ethical limitations prevented us from extending the treatment beyond 4 weeks. In future studies (i.e., IND enabling), we will consider placing a catheter access port in the animals for longer administration schedules. We added these considerations in the Discussion.

C.2. It has been previously shown that Musashi 2 is upregulated in BC CML and drives BC transformation. It is surprising that the authors did not mention MSI2. What is the relationship between MSI2 and miR-142? Does MSI2 suppress miR-142? Does miR-142 suppress MSI2? Or they are unrelated? Also, there are multiple published MSI2 CLIP datasets in hematopoietic cells that can be analyzed to see if MSI2 binds to the miR-142 region.

R: Upregulation of MSI2 has been associated with aggressive CML phenotype (i.e., BC). Thus, the Reviewer's question is pertinent, and we are thankful that it was raised. Since we already provide a relatively large amount of data, a deeper assessment of the interplay between miR-142 and MSI2 would add complexity to the presented model and is beyond the scope of this manuscript. Nevertheless, we showed that increased levels of MSI2 occurred in *miR-142^{-/-}BCR-ABL* LSKs thereby further supporting that BC transformation had occurred.

In detail, to address the Reviewer's question, we have first mined both TargetScan (8.0) and miRDB databases and discovered that MSI2 is a predicted target of miR-142-3p. Based on TargetScan, there is a conserved 8mer miR-142-3p-binding site on the 3' UTR of MSI2 (see Table, also shown in Supplementary Fig. 3i).

Table. The binding site of miR-142-3p to the 3' UTR of MSI2 is conserved in human and mouse:

	Predicted consequential pairing of target region (top) and miRNA (bottom)	Site type	Context++ score	Context++ score percentile	Weighted context++ score	Conserved branch length	P _{CT}
Position 4343-4350 of MSI2 3' UTR hsa-miR-142-3p.1	<pre> 5' ...CCGUCUGGGCACAGACUACA... 3' AGGUUUUUAUCCUUUGUGAUGU </pre>	8mer	-0.21	85	-0.03	3.419	0.68
Position 4075-4082 of MSI2 3' UTR hsa-miR-142-3p.2	<pre> 5' ...AUUGUUUACAAAAGAACACUAA... 3' AGGUUUUUAUCCUUUGUGAUG </pre>	8mer	-0.20	92	-0.03	4.236	0.60

	Predicted consequential pairing of target region (top) and miRNA (bottom)	Site type	Context++ score	Context++ score percentile	Weighted context++ score	Conserved branch length	P _{CT}
Position 4442-4449 of MSI2 3' UTR mmu-miR-142a-3p.1	5' ...CAGUCCUGGGCACAGACACUACA... 3' AGGUUUUUAUCCUUUGUGAUGU	8mer	-0.23	82	-0.04	3.419	0.68
Position 4145-4152 of MSI2 3' UTR mmu-miR-142a-3p.2	5' ...AUUGUUUACAAAAGAACACUAA... 3' AGGUUUUUAUCCUUUGUGAUG	8mer	-0.18	89	-0.03	4.236	0.60

Accordingly, we observed higher MSI2 mRNA and protein levels in LSKs from *miR-142^{-/-}BCR-ABL* (BC CML) mice compared with those from *miR-142^{+/+}BCR-ABL* (CP CML) mice (Supplementary Fig. 3a-b), and in LSKs from *miR-142^{-/-}* mice compared with those from *miR-142^{+/+}* non-leukemic mice (Supplementary Fig. 3c-d). Downregulation of miR-142 in *miR-142^{+/+}BCR-ABL* LSKs with a miR-142 inhibitor (CpG-**anti**-miR-142) was associated with increased MSI2 mRNA and protein levels (Supplementary Fig. 3e-f). Conversely, upregulation of miR-142 levels in *miR-142^{-/-}BCR-ABL* LSKs with miR-142 mimics (CpG-**M**-miR-142) reduced MSI2 mRNA and protein levels (Supplementary Fig. 3g-h).

As suggested by the Reviewer, we also mined previously published MSI2 CLIP datasets (PMC6177596; PMC4857000; and PMC4643281), but found no evidence of miR-142 among the MSI2 targets.

We added these data in the revised manuscript (see page 7-8 and Supplementary Fig. 3a-i).

C.3. A substantial part of the manuscript is describing the metabolic and mitochondrial changes in miR-142^{-/-} CML. I can see the attempt by the authors to address potential mechanisms without identifying functional miR-142 targets. I recognize that it is not always feasible to identify a single target of a miRNA that explains most of its phenotypes. However, this descriptive part is too long and to some degree dilutes the more interesting messages on the biology of miR-142. I suggest the authors shorten these sections.

R: We appreciate the Reviewer's comment and we have shortened this part in the revised manuscript.

Minor comments:

1. For the single cell RNAseq analysis, it seems that the authors do not have data from more than one sample for WT vs miR-142 KO. The population size changes (Fig 2g, 2h) should therefore be only suggestive. Additionally, I am concerned about calling these sub populations from LSKs as HSCs, because there is no evidence that they all have stem cell properties. Calling them progenitors will be more accurate.

R: Samples for single cell RNA-seq analysis were obtained by pooling LSK cells from three miR-142 WT CML mice and three miR-142 KO CML mice, in order to have sufficient material and representation. We clarified this approach in the methods. Regarding the LSK subpopulations, we now call them "LSK clusters", and not HSCs. In the revised paper, we are also referring to these cells as LSKs or hematopoietic cell stem and progenitors (HSPCs) or HSPC- or LSC-enriched populations, instead of HSCs/LSCs.

2. Figures 3h and 3i are misleading. It is a given that when comparing differentially expressed pathways vs the rest of the transcriptome, the per-gene information content will be higher in the pathway. This is because the rest of the transcriptome contains many unrelated or housekeeping genes. These data do not support that EMHGS are necessarily more important than other genes.

R: The reviewer makes a good point regarding differential expression and MI and provides us with an opportunity for additional clarification.

We maintain that the comparison with full transcriptome is appropriate as it shows that the EMHGS contains higher per-gene information density. To address the Reviewer's concern, we have now added a comparison of the number of differentially expressed genes (DEGs) contained in the EMHGS vs those contained in the full transcriptome minus the EMHGS genes and shown that DEGs were not overly represented in EMHGS (hypergeometric p-value = 0.69), placing EMHGS and the full transcriptome at the same footing.

The rationale for comparing the EMHGS to the full transcriptome is two-fold: 1) the EMHGS were identified using GSEA which uses the log-fold change of all genes in the transcriptome – not DEG enrichment; and 2) the MI analysis was performed both to support the correlation analysis in Figure 3g and to understand how a similar state-space could be constructed using a greatly reduced number of pertinent genes, i.e., EMHGS.

We are not claiming that the EMHGS is more important than the full transcriptome in defining CML state transition, rather we say that taken altogether, EMHGS enrichment, state-space, and higher per-gene mutual information density support that the EMHGS is sufficient to encode the CML state and, therefore, support the biological relevance of EMHGS to the transition from CP to BC, as we also demonstrated using metabolomic profiling and functional assays.

To make these points clearer, we have modified the manuscript as follows:

In the old version: Thus, changes in the EMHGS alone as induced by the acquisition of miR-142 deficit were able to inform on the state transition from CP- to BC-LSCs better than the changes in the remaining transcriptomes, thereby supporting the biological relevance of these genes to BC transformation.

In the new version: Thus, we concluded that expression changes of EMHGS as induced by the acquisition of miR-142 deficit, contained sufficient information to capture the state-transition of BCR/ABL LSKs from CP to BC, thereby supporting the biological relevance of the EMHGS changes to the disease evolution.

3. Fig 4b: the annotation of significance (the *) is too small, and hard to see.

R: We corrected it.

4. Fig 6b: Please describe what red and blue lines are. If red is scrambled mimic, why there is no uptake of the scrambled?

R: We labeled the red and blue lines in Fig 6b in the revised manuscript. The red line represents a control treated with vehicle and accordingly shows no uptake.

Reviewer #3:

Overall the authors apply sc-RNA-seq, various types of metabolomic readouts and electron microscopy on a new mice model of miR-142 deficient CML and patient derived samples. For the latter, quite a large amount of work has been done with rare human LSCs (CD34+CD38-), which is impressive. The paper is well written, and the topic is of interest to the reader of Nature Communications.

R: We thank the Reviewer for the positive comments.

The authors correlated the increase in metabolic activity following miR-142 depletion with an upregulation of Mitofusin-1 (MFN1), enhanced mitochondrial fusion, decreased autophagy flux (LC3 levels) and decreased ROS levels. This part is preliminary/underdeveloped and mainly speculative. Further experiments are therefore required, with increased n-numbers and statistical analysis, to strengthen this part (i.e., while the electron microscopy looks clear, quantification or multiple representative images would support this).

R: We appreciate the Reviewer's comments and have addressed them along with the other Reviewers' comments throughout the revised manuscript.

Specific comments:

C.1. Figure 1 and Supplementary related figures 1-2: What are the absolute counts here for all cell types in spleen and bone marrow, based on total number of cells (cell counts). Maybe LT-HSCs are getting exhausted or if number of LSKs is high enough, a reduced % LT-HSCs won't translate to reduced total number.

R: We have now updated the figures to provide the absolute counts of all cell types in the BM (per femur) and spleen (see page 6, and Supplementary Figures 1b-c).

C.2. Metabolomic analysis on page 12: How is the replicates plotted obtained if samples are pooled? How were they pooled, was it 2 mice to one sample or were all mice pooled and technical replicates made?

R: Since at least 3×10^6 cells per sample were necessary for metabolomic studies and we could not obtain a sufficient number of BM Lin^c-Kit⁺ cells from individual *miR-142*^{-/-}*BCR-ABL* or *miR-142*^{+/+}*BCR-ABL* mice, we randomly combined BM Lin^c-Kit⁺ cells from 2-3 individual mice to reach the number of 3×10^6 cells per sample. So, we obtained a total of 10 samples from 24 *miR-142*^{-/-}*BCR-ABL* mice and 8 samples from 18 *miR-142*^{+/+}*BCR-ABL* mice, which were then analyzed for metabolomics. We clarified this in the method of the revised manuscript.

C.3. What pathway analysis was used for untargeted metabolomics?

R: We did not employ enrichment-based pathway analysis tools to mine our unbiased metabolomics dataset. Instead, we mapped metabolite classes to classical metabolic pathways (i.e., SMPD and KEGG).

C.4. Figure 5b: Figure legend suggests n=3. Quantification with statistical analysis would be useful.

R: Done

C.5. Figure 5d-e: The Seahorse experiments require further explanation in text/figure legends. Why have the authors used Oligomycin and FCCP simultaneously? Here it is important to include antimycin/rotenone to shut down OCR and ascertain the differences in basal and maximal OCR are not simply due to different cell numbers. Statistical analysis is also required when interpreting the data.

R: See response for Reviewer #1, C.4 and new Figure 5d-e and Supplementary Figure 8g-j.

C.6. Figure 5f-k: Quantification and statistical analysis are required to interpret these data.

R: Done (see Supplementary Figures 9g, 9i, and 9m).

C.7. Supplemental Figure 6: n=2 or not stated for western blots (this should be in legends for all figures in general). Quantification and statistical analysis are required for key experiments.

R: Done

C.8. Supplementary Figure 7h-i, I: Data regarding autophagy are not convincing. Is this LC3-I or lipidated LC3-II? Here autophagy flux should be measured using appropriate inhibitors (i.e. Bafilomycin) in at least one of the models (mouse or human). The levels of LC3 by Western alone cannot accurately predict autophagy flux (the authors may consider measuring LC3 puncta by IF).

R: Please see also response to **Reviewer 1 C.6**. To detect the levels of LC3, we employed LC3 antibody from Sigma (#L8918) and Novus Biological (#NB100-2220). According to the molecular weight on western gel, these bands represented LC3-II. As suggested by Reviewers #1 and #3, we also performed IF staining for mouse miR-142^{-/-} B/A LSK and human BC CD34⁺38⁻ cells untreated and treated with SCR control or M-miR-142 using LC3 antibody. We showed that higher levels of miR-142 levels associates with LC3 puncta formation (see Figure 1 in Response to Reviewer 1 C.6.).

The Reviewer is absolutely right; to correctly interpret the LC3 changes, a lysosomal inhibitor (e.g., bafilomycin A1) should also be used. If the increase in LC3 observed after the treatment of interest (i.e., M-miR-142) is maintained following exposure to bafilomycin A1, one may reasonably conclude that the treatment has changed the autophagy flux. Conversely, it may be a false positive (Klionsky et al., *Autophagy*, 2016; Gump et al., *Autophagy*, 2014). Herein, we observed that bafilomycin A1 did not reverse the increased levels of LC3 observed after M-miR-142 treatment (see Figure 1 in Response to Reviewer 1 C.6), suggesting that M-miR-142 had “truly” increased autophagy.

Nevertheless, given that this aspect is tangential to the main point of the manuscript and may require additional studies, and because of the already existing complexity of the proposed model and reached space limitations, we have now elected to eliminate this part and to pursue this line of research in the future. Of course, we will be happy to restore it if the Reviewers and the Editors wish so.

C.9. Supplementary Figure 7j: JC-1 experiment is not described well in the text or figure legends. Also, quantification and statistical analysis is required.

R: We reworded the description of the JC-1 experiment in Figure legend and added quantification and statistical analysis in the revised manuscript. (See Supplementary Figure 9k-l and legends)

C.10. Figure 7: Are same samples used in Figure 6 h-l and 7 e-f (i.e., what is the effect on Nilotinib alone in figure 7 e-f)?

R: The same patient sample was used to engraft the mice in Figure 6 h-l and 7 e-f, but the two experiments were run separately using different cohorts of mice (as they became available from our breeding program).

References: There has been limited work done on miR-142 in CML which should be referenced (for example: <https://www.frontiersin.org/articles/10.3389/fonc.2021.718731/full>). The authors cite the following paper (<https://www.jci.org/articles/view/123839>) but don't discuss that in dendritic cells, where miR-142 also regulates FAO. It would be useful to mention this a potential effect on normal or abnormal immune function.

R: We now referenced this additional work including the effect of miR-142 deficit on dendritic cells' activation (see page 22-23). We also have evidence from our own studies that, in addition to myeloid hematopoietic progenitors, distinct immune cell populations acquire miR-142 deficit during BC transformation, become dysfunctional and may contribute to the disease evolution. To this end, a second manuscript is currently being written. We plan to submit this paper to Nature Comm. as part two of our work.

Graphical abstract is underdeveloped and should be improved.

R: Done

REVIEWERS' COMMENTS

Reviewer #1 (Remarks to the Author):

The authors have addressed my concerns.

Reviewer #3 (Remarks to the Author):

The authors have addressed all my concerns.

Regarding my comment on the autophagy part (point 8 – also commented on my Reviewer 1, point 6), which the authors have addressed in the Rebuttal letter, I would like to mention that the treatment with bafilomycin A1 should be carefully interpreted. If M-miR 142 treatment is inducing autophagy flux, one would expect a further increase in LC3 puncta when the flux is inhibited with bafilomycin. Given that bafilomycin does not lead to further increase in LC3 puncta, it is still possible that M-miR 142 treatment is actually inhibiting autophagy flux, rather than inducing it. Therefore, I agree with the authors that additional experiments are necessary to fully address this point and support their decision to remove the data from the manuscript as this does not affect the main message of the paper.

Reviewer #4 (Remarks to the Author):

The authors have responded to all the comments and performed a significant set of experiments that strengthen the manuscript.

RESPONSE TO REVIEWERS' COMMENTS

Reviewers' comments:

Reviewer # 1 (Remarks to the Author):

C: The authors have addressed my concerns.

R: We thank the reviewer for the careful evaluation of our manuscript and are happy to know that the Reviewer is satisfied with the revised paper.

Reviewer #3 (Remarks to the Author):

C: The authors have addressed all my concerns.

Regarding my comment on the autophagy part (point 8 – also commented on my Reviewer 1, point 6), which the authors have addressed in the Rebuttal letter, I would like to mention that the treatment with bafilomycin A1 should be carefully interpreted. If M-miR 142 treatment is inducing autophagy flux, one would expect a further increase in LC3 puncta when the flux is inhibited with bafilomycin. Given that bafilomycin does not lead to further increase in LC3 puncta, it is still possible that M-miR 142 treatment is actually inhibiting autophagy flux, rather than inducing it. Therefore, I agree with the authors that additional experiments are necessary to fully address this point and support their decision to remove the data from the manuscript as this does not affect the main message of the paper.

R: We thank the Reviewer for agreeing with our decision and are happy to know the Reviewer is satisfied with the revised paper.

Reviewer #4 (Remarks to the Author):

C: The authors have responded to all the comments and performed a significant set of experiments that strengthen the manuscript.

R: We thank the reviewer for the careful evaluation of our manuscript and are happy to know that the Reviewer is satisfied with the revised paper.